# RAWDet-7: A Multi-Scenario Benchmark for Object Detection and Description on Quantized RAW Images

## Abstract

Most vision models operate on 8-bit standard RGB (sRGB) images produced by dedicated image sensor processing pipelines designed for human perception rather than machine reasoning. In contrast, RAW images preserve sensor measurements, dynamic range, and fine-grained scene structure that can be critical for downstream understanding. Yet, progress in the RAW-domain vision remains limited by the lack of large-scale, high-quality benchmarks. To close this gap, we introduce RAWDet-7, a multi-scenario benchmark for object detection and object description on quantized RAW images, comprising ∼25k training and 7.6k test images consolidated from four datasets across diverse cameras, sensors, bit-depth precision, lighting conditions, and environments. RAWDet-7 provides dense, standardized annotations for seven object categories, correcting missing labels, hallucinations, and inconsistencies in prior datasets, especially for small and partially-occluded instances. Beyond detection, we introduce an object-description track with detailed object-level descriptions derived from high-resolution sRGB references, enabling the study of how well different RAW processing pipelines remain aligned with high-resolution sRGB reference descriptions for fine-grained semantic, spatial, and contextual information. Finally, RAWDet-7 supports controlled benchmarking under simulated 4-bit, 6-bit, and 8-bit quantization. Across standard detectors and a large grounding model, we show that suitable RAW-aware input mappings make low-bit RAW on par with sRGB pipelines, and, in the object-description track, achieve substantially higher agreement with high-resolution sRGB references than naïve quantization. Dataset & code upon acceptance.

## 1 Introduction

Modern vision systems are typically built and evaluated on standard RGB (sRGB) images at 8-bit precision. At capture time, however, image sensors record the scene in a substantially higher bit-depth (∼24-bit) RAW format, which is then mapped to sRGB by an image signal processor (ISP). This mapping consists of multiple stages, including black-level subtraction, demosaicking, denoising, white balancing, gamma correction, color correction, and compression, and is primarily designed to produce visually pleasing images for humans. However, this photography-oriented pipeline is not optimized for downstream machine perception.

Compared to processed sRGB images, RAW images retain more faithful sensor measurements, including the original bit depth and dynamic range. Leveraging RAW data can therefore benefit downstream vision tasks (Omid-Zohoor et al., 2014; Xu et al., 2023). At the same time, RAW data varies substantially across sensors, which differ in bit depth, response characteristics, and ISP design. For specialized applications (Klinghoffer et al., 2022; Sommerhoff et al., 2023; Agarwal et al., 2025), jointly optimizing sensing, image processing, and the downstream task can therefore be highly beneficial. Despite this promise, progress in RAW-domain vision and end-to-end learned task-specific imaging remains limited by the lack of large-scale, well-annotated, and standardized RAW benchmarks.

To address this limitation, we introduce RAWDet-7, a multi-scenario benchmark for object detection and object description on RAW images. We focus on object detection because it requires both recognition and precise localization of class-specific visual evidence. In addition, we introduce an object-description track

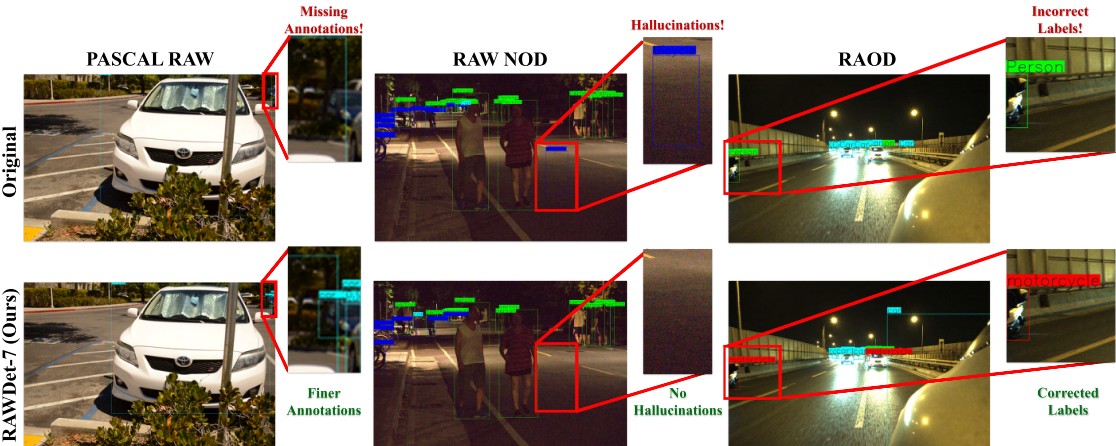

Figure 1: Comparing ground truth annotations provided in the original datasets and the new ones proposed in RAWDET-7. Our proposed annotations are more fine-grained, as seen for PASCAL RAW, which originally annotated only one instance of the cars in the image; we annotate all the other instances of cars in the image (with a 20% overlap threshold i.e. at least 20% area of the bounding box should be non-overlapping). RAW NOD (-Nikon and -Sony), RAOD (-Day and -Night) original annotations contain hallucinations as seen here for RAW NOD, which hallucinates a bicycle in the center right of the frame. Original annotations even contain misclassifications, as seen for RAOD, which misclassified a motorcycle as a person. RAWDET-7 **(bottom) overcomes these drawbacks.**

on a curated subset of images to study agreement with high-resolution sRGB reference descriptions after RAW preprocessing and low-bit quantization for fine-grained semantic, spatial, and contextual information. The proposed RAWDET-7 is larger than previous RAW detection datasets, comprising ∼25k training and ∼7.6k test images obtained by consolidating four different datasets (Omid-Zohoor et al., 2014; Xu et al., 2023; Morawski et al., 2022; Ignatov et al., 2020) across varied lighting conditions, camera types, and environments.

Many prior RAW datasets cover only a small number of classes and often annotate only large, prominent objects. We therefore relabel the consolidated dataset using a large foundation model, followed by human validation on the corresponding sRGB images, to obtain annotations for 7 object categories aligned with the community-accepted naming conventions of MS-COCO (Lin et al., 2014) and LVIS (Gupta et al., 2019). This produces substantially denser annotations, both by increasing category coverage and by improving the labeling of small and partially-occluded instances, thereby making RAWDET-7 more useful for developing robust detection models that generalize across real-world scenarios. As illustrated in Fig. 1, RAWDET-7 provides significantly richer and corrected annotations than prior datasets for the same images. Since annotation quality is central to the value of the benchmark, we further validate the reliability of the relabeling pipeline through human studies, confirming that the resulting annotations are consistently preferred over the previously existing ones.

While RAWDET-7 contains full-precision RAW images, reducing bit depth can lower memory usage and provide significant power savings (Yin et al., 2021). However, there is still limited understanding of how complex downstream vision models behave under heavily quantized RAW input. We therefore benchmark object detection performance under simulated low-bit quantization, in addition to standard sRGB input. Specifically, we consider low-bit quantized inputs obtained using different information processing methods: linear scaling, logarithmic scaling (Buckler et al., 2017), learnable $\gamma$ scaling (Ljungbergh et al., 2023), and a combination of logarithmic and learnable $\gamma$ scaling (Fatima et al., 2025).

Through systematic benchmarking on low-bit quantized RAW inputs, we show that models trained on RAWDET-7 outperform counterparts trained on the individual source subsets across quantization levels, with more robust detections for small and occluded instances, further demonstrating the value of the benchmark. Moreover, essentially all prominent object detection methods considered here, including their architectural and preprocessing design choices, were developed for sRGB input. Despite this clear bias, our

experiments show that object detectors trained on RAW input, especially low-bit RAW input with suitable input scaling, can perform on par with models trained on the same amount of sRGB data.

Beyond detection, we evaluate how closely object descriptions from different RAW processing pipelines agree with high-resolution sRGB reference descriptions for a large vision-language model trained primarily on sRGB images. To this end, we use detailed descriptions generated from high-resolution sRGB images as reference descriptions, and compare them against descriptions obtained from different processed RAW variants of the same content. This setup does not treat sRGB captions as absolute semantic ground truth; rather, it measures agreement with high-resolution sRGB-Gemini reference descriptions, reflecting how much object-level information remains accessible to an sRGB-biased model after RAW preprocessing. We further validate this object-description pipeline through human studies, showing that the generated descriptions are of high quality and that LLM-based judgments are meaningfully aligned with human ratings. Our results show that descriptions generated from suitably processed RAW images have substantially higher agreement with the high-resolution sRGB references and provide more accurate and detailed object explanations than descriptions obtained from naïvely processed low-bit RAW images.

The contributions of this work are as follows:

- **A consolidated RAW benchmark.** We introduce RAWDET-7, a large-scale benchmark for object detection and object description on quantized RAW images, built by consolidating four datasets and re-annotating them into seven standardized categories with denser labels, especially for *small* and *partially-occluded* objects. Thereby, the object-description track allows assessing agreement with high-resolution sRGB reference descriptions for fine-grained semantic, spatial, and contextual information.

- **Human-validated benchmark quality.** We validate the benchmark with human studies both for detection and description annotations. Users prefer our consolidated detection annotations over the original ones and confirm the quality of the object-description annotations.

- **Controlled low-bit RAW benchmarking.** We benchmark downstream vision under simulated 4-bit, 6-bit, and 8-bit RAW quantization with four input scaling strategies, and show that suitable RAW-aware mappings make low-bit RAW competitive with standard sRGB pipelines. We show that these findings often hold across models and tasks, particularly between conventional object detectors and a recent large grounding model.

## 2 Related Work

In the following, we discuss object detection and description, RAW-domain detection, and RAW imaging datasets and restoration relevant to the proposed RAWDET-7. Broader sensor-to-task and restoration studies include Dirty Pixels (Diamond et al., 2021), which jointly learns RAW processing and recognition, DeepISP (Schwartz et al., 2019), which learns an end-to-end low-light image-processing pipeline, and Reconfiguring the Imaging Pipeline for Computer Vision (Buckler et al., 2017), which studies task-oriented low-precision sensing.

**Object Detection.** Deep object detection spans two-stage methods such as **Faster R-CNN** (Ren et al., 2016), which generates region proposals before classification and refinement; one-stage methods such as **RetinaNet** (Lin et al., 2017) and **PAA** (Kim and Lee, 2020), which perform detection in a single pass and respectively use focal loss and probabilistic anchor assignment; and open-set grounding models such as **MM-Grounding-DINO** (Zhao et al., 2024), which localizes objects from language supervision beyond fixed label spaces. We benchmark these representative paradigms to test whether observations on RAW and quantized RAW inputs generalize across conventional and large-scale detectors. Since these detectors are primarily developed for sRGB, RAW-domain work adapts the ISP, input, or detector. ISP-Teacher (Zhang et al., 2024) performs ISP-aware domain adaptation for dark object detection, RAW-Adapter (Cui and Harada, 2024) introduces learnable input- and model-level adapters, Raw or Cooked? (Ljungbergh et al., 2023) and Learnable CCM (Liu et al., 2024a) couple simplified RAW or inverse-ISP mappings with detector training, and Dense Object Detection in RAW UAV Imagery (Wu et al., 2024) applies YOLOv8 directly to

RAW inputs. Hardware-in-the-Loop ISP Optimization (Mosleh et al., 2020) and End-to-End HDR Camera Pipeline Optimization (Robidoux et al., 2021) jointly optimize ISP parameters and detection, while DynamicISP (Yoshimura et al., 2023a) dynamically controls classical ISP parameters, Rawgment (Yoshimura et al., 2023b) provides noise-accounted RAW augmentation, and AdaptiveISP (Wang et al., 2024) learns scene-adaptive ISP structures and parameters. AODRaw (Li et al., 2025) combines diverse-condition benchmarking with RAW-domain pre-training, SFAE (Ye et al., 2025) fuses spatial and frequency representations to recover suppressed object details, and SimROD (Xie et al., 2026) uses lightweight global-gamma and local green-channel enhancement. These approaches optimize particular data-generation, ISP, or detector pipelines rather than providing a unified, annotation-consistent benchmark across sensors and bit depths.

**Object Description.** Recent work uses set-of-mark and visual prompting for region-level grounding and object-centric captioning in large vision-language models. Set-of-Mark Prompting (Yang et al., 2023a) overlays alphanumeric marks and segmentation-derived regions so that a VLM can reason about specific objects and their relations. Contrastive Region Guidance (Wan et al., 2024) compares predictions on highlighted and blacked-out regions to reduce prior biases and better follow region prompts. Scaffolding Coordinates (Lei et al., 2025) adds an image-coordinate grid and textual references to improve spatial reasoning and vision-language coordination. ViP-LLaVA (Cai et al., 2024) trains multimodal models to interpret userdrawn boxes, arrows, and scribbles as region selectors, while Fine-Grained Visual Prompting (Yang et al., 2023b) uses precise segmentation masks and blur-outside-mask operations to focus on a target instance. In RAWDET-7, we adopt a set-of-marks interface and additionally provide a high-level object class for each numeric mark when prompting the VLM, allowing us to study how well detailed object descriptions agree with high-resolution sRGB references across different quantizations.

**RAW Imaging Datasets and Restoration.** Most existing object-detection datasets and models use sRGB images produced by ISP pipelines optimized for human perception, whereas RAW images preserve richer scene information and higher dynamic range that may benefit detection. However, large-scale annotated RAW datasets remain scarce. Existing resources include PASCAL RAW (Omid-Zohoor et al., 2014), NOD/GenISP (Morawski et al., 2022), and RAOD (Xu et al., 2023). Efficient Visual Computing with Camera RAW Snapshots (Li et al., 2024a;b) additionally generates simulated RAW data through an inverse ISP and releases MultiRAW for RAW-domain detection. Collectively, these datasets remain limited in scale, object classes, conditions, or annotation quality: PASCAL RAW annotates mainly large objects, NOD and RAOD contain hallucinated or noisy labels, and all three cover a narrow range of conditions. RAOD also evaluates reduced-bit RAW detection, but does not provide an annotation-consistent benchmark spanning controlled bit depths. Complementary restoration work is closely tied to paired RAW dataset construction. Learning to See in the Dark (Chen et al., 2018) pioneered end-to-end low-light RAW-to-sRGB restoration using paired short- and long-exposure images. PyNET (Ignatov et al., 2020) replaces a mobile RAW-to-RGB ISP with an end-to-end restoration network and introduces the Zurich paired dataset, which lacks detection labels and is re-annotated in our work. RAISE (Dang-Nguyen et al., 2015) instead targets digital-image forensics rather than object detection. In contrast, our proposed dataset, RAWDET-7, unifies and re-annotates existing datasets with fine-grained labels for seven categories across varied lighting conditions, sensor types, and scenes, and enables controlled benchmarking under simulated 4-bit, 6-bit, and 8-bit quantization.

## 3 Proposed Dataset RAWDET-7

RAWDET-7 provides high quality annotations for object detection as well as for object description. The following sections provide the details of the annotation process.

### 3.1 Object Detection Annotations

As discussed in Sec. 1, progress in RAW-domain vision has been limited by the lack of large-scale, high-quality benchmarks and by the practical challenges of handling RAW data. Although several RAW object detection datasets (Omid-Zohoor et al., 2014; Morawski et al., 2022; Xu et al., 2023) have been proposed, we discussed in Sec. 2 that they each suffer from practical limitations. For example, PASCAL RAW (Omid-Zohoor et al., 2014) and NOD (Morawski et al., 2022) focus on only a few coarse object classes, restricting annotations largely to 'Car', 'Person', and 'Bicycle'. As shown in Fig. 1, several annotations are missing

Table 1: Comprehensive comparison of datasets in RAWDET-7 and other datasets.

| Dataset | Sensor Model | Resolution | Bit Depth | Scenarios | Train Images | Test Images | # Classes | # Sensors |
|---|---|---|---|---|---|---|---|---|
| PASCAL RAW (Omid-Zoohor et al., 2014) | Nikon D3200 DSLR | 6034 × 4012 | 12 | Day | 2128 | 2130 | 3 | 1 |
| Zurich (Ignatov et al., 2020) | MP Sony Exmor IMX380 | 2944 × 3958 | 10 | Day | 46.8k | 1.2k | - | 1 |
| Raw-NOD-Nikon (Morawski et al., 2022) | Nikon D750 | 3968 × 2640 | 14 | Night | 3373 | 843 | 3 | 1 |
| Raw-NOD-Sony (Morawski et al., 2022) | Sony RX100 VII | 4256 × 2848 | 14 | Night | 3309 | 828 | 3 | 1 |
| RAOD (Xu et al., 2023) | Sony IMX490 | 2880 × 1856 | 24 | Day & Night | 16089 | 4000 | 5 | 1 |
| MultiRAW (Li et al., 2024b) | Multiple | Various | 10, 12, 14, 24 | Day & Night | 5154 | 2315 | 10 | 5 |
| RAWDET-7 **(Ours)** | Multiple | Various | 10, 12, 14, 24 | Day & Night | **24864** | **7688** | 7 | 5 |

even within these classes, particularly for small or partially occluded people and vehicles. Existing datasets also contain annotation errors and hallucinations, e.g., in Fig. 1, a motorcycle (under the original 'vehicle' category) from RAOD (Xu et al., 2023) is mislabeled as a 'Person'. Beyond annotation quality, sensor-specific differences in bit depth, sensor characteristics, and lighting conditions introduce substantial variability that is only partially captured by previous datasets. MultiRAW (Li et al., 2024b) attempts to broaden class and scenario coverage, but introduces contradictory categories such as 'rider' and 'person' while still remaining too small overall for a stable and standardized benchmark.

To address these limitations, we introduce RAWDET-7, a comprehensive benchmark dataset for object detection on RAW images. Comprising more than 32k RAW images, with ∼25k for training and 7.6k for testing, RAWDET-7 consolidates and standardizes four existing datasets: PASCAL RAW (Omid-Zoohor et al., 2014) (12-bit), RAOD (Xu et al., 2023) (24-bit, including day and night HDR scenes), Zurich RAW (Ignatov et al., 2020) (10-bit), and NOD (Morawski et al., 2022) (14-bit, with Sony and Nikon sensors). For the Zurich RAW dataset, the majority of the provided images are 488×488 patches randomly cropped from images (not provided otherwise as full scene images in the dataset); we therefore retained only the full-resolution images for which both RAW and sRGB pairs were available. Additionally, since the Zurich dataset is used for RAW-to-sRGB conversion rather than object detection, no class annotations are available, as shown in Table 1 Together, these sources cover a wide range of imaging conditions, including daytime, nighttime, and HDR scenes, as well as diverse camera hardware and sensor bit depths. By unifying them under a consistent annotation scheme, RAWDET-7 enables controlled studies of generalization across lighting conditions, scene complexity, and sensor variability. Table 1 reflects the splits as provided in the various datasets, while the train and test splits used within RAWDet-7 are provided in Figure 16 in Appendix Section H.

A central contribution of RAWDET-7 is its improved and standardized labeling. To correct missing annotations, hallucinations, and inconsistencies in the source datasets while establishing a uniform labeling protocol, we re-annotated all datasets using Grounded-DINO 1.5 (Liu et al., 2024b) (paid API version at: `https://deepdataspace.com/request_api`) for seven object categories: *'Car'*, *'Truck'*, *'Tram'*, *'Person'*, *'Bicycle'*, *'Motorcycle'*, and *'Bus'*, following MS-COCO and LVIS-style naming conventions. Grounded-DINO 1.5 was run exclusively on sRGB images. For all datasets except RAOD, the sRGB images provided by the original dataset authors, processed through their respective ISPs, were used as input. For RAOD, where no sRGB images were publicly available, we synthesised them from the RAW data using our own ISP pipeline, described in detail in Section G. Since sRGB and RAW images do not always share the same resolution, bounding boxes were transferred to RAW-domain coordinates by rescaling according to the ratio between the RAW and sRGB image resolutions. We then refined the generated annotations by retaining only confident predictions above a confidence threshold of 0.8. This threshold was chosen after manual inspection of annotation quality and accuracy over a large subset of RAWDET-7. As a result, RAWDET-7 provides denser and more consistent annotations across the consolidated dataset, with substantially improved coverage of small and occluded object instances. The improvement is not only qualitative: in an annotation-quality user study with 53 participants and 1325 pairwise comparisons, users preferred the RAWDET-7 annotations over the original annotations in 1002 cases, i.e., 75.62% of all comparisons. Please note that post hoc human audits of model-assisted annotations may be subject to confirmation bias, as auditors can overlook the same objects missed by the model.

## 3.2 Object Description

While object detection provides precise localization, it offers only a coarse, class-level characterization of scene content. To evaluate how RAW preprocessing affects object-level information accessible to an RGB-pretrained vision-language model, especially for the annotated object categories of interest, we introduce the task of *object description*. The goal is not to define absolute semantic ground truth or absolute information

preservation, but to measure agreement with high-resolution sRGB-Gemini reference descriptions generated under the same set-of-marks protocol and in particular the level of detail therein.

To this end, we generate high-quality reference descriptions using Gemini-2.5-Pro (Comanici et al., 2025) for objects localized by the ground-truth bounding boxes in RAWDET-7. Specifically, for a curated subset of 500 high-resolution sRGB images, we overlay visual marks on annotated objects following the set-of-marks procedure of (Yang et al., 2023a), using black square markers with colored numbers indicating object IDs. We then prompt Gemini-2.5-Pro to produce detailed descriptions for each marked object. To account for output variability and to establish a reference agreement level, we prompt the model twice per image. Our prompts, qualitative examples, and human-study validation are provided in Sec. J, with the caption-quality user study in Sec. J.2 and Fig. 17, where 63 participants and 1260 ratings assign the generated object descriptions an average score of $4.07 \pm 0.48$ out of 5, i.e., approximately 82%, indicating that the reference descriptions are of high quality even under human evaluation.

For evaluation, we compare descriptions generated from the high-resolution sRGB images against those generated from downsampled sRGB variants and differently processed RAW variants at multiple bit depths, using the same set-of-marks. We report Regex Match overlap and BLEU score, and additionally use Gemini-2.5-Flash-Lite as a judge to score semantic similarity, level of detail, and precise detail match between the reference sRGB descriptions and those from each variant. Thus, these metrics measure agreement with high-resolution sRGB-Gemini descriptions, not absolute semantic correctness; the judge protocol is validated against humans in Fig. 19 and with Claude Sonnet 4.6 in Sec. J.6.

## 4 Scaling Methods Used For Benchmarking

One key challenge with using RAW images in machine learning is that their high precision and large size, make them computationally expensive to store and process. Quantization offers a practical solution by reducing bit depth, but it introduces new design choices around how to scale inputs before quantization. As an example use-case, in this section, we explore combinations of quantization with known input scaling methods, such as logarithmic and gamma mappings, to enable efficient and effective object detection directly on low-bit RAW inputs. Recording at lower bit-depth and skipping the ISP before feeding images to a downstream task model can provide power and memory gains (Yin et al., 2021), making such methods practical in scenarios like low-bandwidth or low-power consumption sensing. It should be noted that our low-bit inputs are obtained via post-hoc quantization of higher-bit RAW data rather than direct capture at reduced bit depth. This procedure simulates low-bit representation but does not replicate the ADC-specific noise characteristics or photon-limited shot noise inherent to real low-bit sensor hardware.

We benchmark the performance of RAWDET-7 using low-bit depth images obtained using the following scaling methods, where $\mathcal{X}$ with bit-depth $N$ is the input to the quantizer $\mathcal{Q}(\cdot)$, which provides output with target bit depth $\hat{N}$:

1. **Linear quantization** converts input into a quantized digital value using uniform linear steps as

$$\mathcal{Q}(\mathcal{X}) = \left\lfloor \mathcal{X}_{\text{norm}} \cdot (2^{\hat{N}} - 1) \right\rfloor, \tag{1}$$

where, $\mathcal{X}_{\text{norm}} = \left(\frac{\mathcal{X}}{2^N - 1}\right)$ is the normalized version of $\mathcal{X}$. This naïve approach of reducing the bit-depth often fails, motivating better methods for scaling.

2. **Logarithmic quantization** proposed by (Buckler et al., 2017; Bermak and Kitchen, 2006) non-linearly scales the dynamic range of $\mathcal{X}$ before quantization,

$$\mathcal{Q}(\mathcal{X}) = \left\lfloor \frac{\mathcal{X}_{\text{log}} - \min(\mathcal{X}_{\text{log}})}{\max(\mathcal{X}_{\text{log}}) - \min(\mathcal{X}_{\text{log}})} \cdot (2^{\hat{N}} - 1) \right\rfloor, \tag{2}$$

where, $\mathcal{X}_{\text{log}} = \log(\mathcal{X} + \epsilon)$, with $\epsilon$ is accounting for the mapping to issue values $\mathcal{X}_{\text{log}} \geq 0$, i.e., (Buckler et al., 2017) use $\epsilon = 1$.

3. **Quantization with Learnable $\gamma$ -Scaling:** $\gamma$ mapping, similar to logarithmic quantization, also compresses the dynamic range of the input, and we consider a learnable $\gamma$ mapping before quantization as

$$\mathcal{Q}(\mathcal{X}, \gamma) = \left\lfloor \mathcal{X}_{\text{norm}}^{\gamma} \cdot \left(2^{\hat{N}} - 1\right) \right\rfloor, \tag{3}$$

where $\mathcal{X}_{\text{norm}}$ is the normalized version of $\mathcal{X}$ as defined earlier in linear quantization, and the parameter $\gamma$ is learned along with the parameters $\theta$ of the neural network. In addition to learning a single $\gamma$ for the dataset, we also evaluate conditioning $\gamma$ on the domain, such as lighting conditions or sensors. $\mathcal{Q}(\mathcal{X}, \gamma)$ is hereby referred to as $\gamma$-SCALING.

4. Since various non-linear mappings of input intensities are an active area of research for the sensors community (Gulve et al., 2023; Lefebvre and Bol, 2025), we benchmark a combination of logarithmic and learned $\gamma$ quantization. That is, we use $\mathcal{X}_{\text{norm}} = \mathcal{X}_{\text{log}}$ in Eq. (3).

### 4.1 Learning Task-specific Low-bit Quantization

We learn the parameter $\gamma$ of $\gamma$-SCALING and 'Log + $\gamma$-SCALING' jointly with the neural network parameters for the specific task by optimizing

$$\min_{\theta, \gamma} \ \mathbb{E}_{(\mathcal{X}, y)} \left[\mathcal{L}(\eta(\mathcal{Q}(\mathcal{X}, \gamma)), y; \theta)\right], \tag{4}$$

where $\eta$ is the task-specific neural network with parameters $\theta$, and where $\mathcal{L}$ is a suitable loss function comparing the network output and the ground truth prediction y.

Since RAWDET-7 offers a unique multi-scenario setting, we also learn scenario-specific $\gamma$ values. Using one $\gamma$ value over the entire dataset is denoted as 1-$\gamma$. When using two $\gamma$ values we denote it as 2-$\gamma$, with the intention of them conditioning to the time of day, i.e. one $\gamma$ for daytime and one for nighttime images. Lastly, we test using five $\gamma$ values, denoted as 5-$\gamma$, allowing one optimized to each sensor.

For optimization, since the quantization operation stops the flow of gradients in the backward pass, we use a straight-through estimator to allow gradient-based optimization of $\gamma$. Please note that $\gamma < 0$ would lead to undesirable behavior, i.e. , $\mathcal{X}_Q \geq 2^{\hat{N}} - 1$ when $N \geq \hat{N}$ (which is the case when we want to quantize to lower bit depths). To avoid this, we clamp $\gamma$ to be non-negative, using ReLU(.) (Glorot et al., 2011). To take into account Bayer-patterned images in the RAW dataset, we extract red, green, and blue channels from the Bayer-patterned image and downsample it using nearest neighbor interpolation to simulate the capture of a low-resolution, low-bit-depth image, which is provided as input to the neural network. This pipeline for joint training of $\gamma$ and task-specific neural network parameters $\theta$ using quantized RAW images is the methodology for the forthcoming evaluations.

## 5 Experiments

For object detection, we use the mmdetection (Chen et al., 2019) framework from openmmlab. Following previous works (Xu et al., 2023; Omid-Zohoor et al., 2014) that perform object detection on RAW images, we conduct experiments using three architectures Faster R-CNN (Ren et al., 2016), RetinaNet (Lin et al., 2017), and PAA (Kim and Lee, 2020) models with a pre-trained ResNet50 (He et al., 2016) backbone.

All training conditions are identical across RAW and sRGB inputs, including augmentation and training budget. For all RAW experiments, the R, G, G, B Bayer channels are extracted and the two green channels are averaged to produce a three-channel representation. For datasets where RAW and sRGB share the same resolution (RAOD, Nikon, and Sony), this results in half the effective spatial resolution of the original RAW input. For Pascal RAW and Zurich, where RAW and sRGB resolutions already differ, the resulting resolution varies accordingly.

We report the following quantitative evaluation metrics: mean average precision (mAP) is calculated by averaging the Average Precision (AP) values across the intersection over union (IoU) thresholds ranging from [0.5, 0.95] with a step size of 0.05, resulting in 10 threshold values. We also report performance

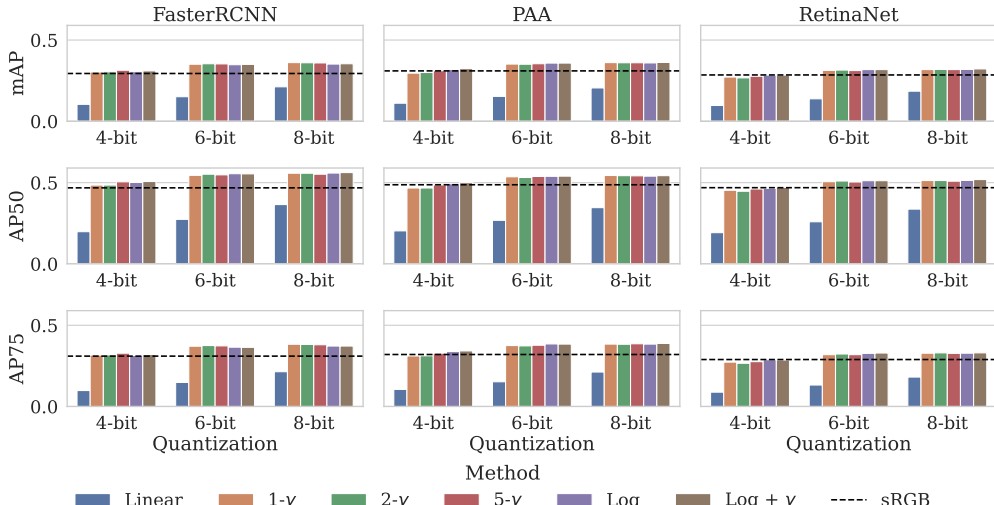

Figure 2: Benchmarking performance on RAWDET-7. Baselines such as logarithmic quantization and jointly learnt $\gamma$ improve results across quantization levels and architectures.

with IoU-thresholds of 0.5, 0.75, and 0.95 (AP50, AP75 and 0.95) for each case. Results with 4 channels (R,G,G,B) are given in Sec. I, see Tab. 4. We use an ImageNet-1k (Russakovsky et al., 2015) pretrained backbone and freeze the first stage of the backbone during training. We follow the multi-scale training setup commonly used (Ren et al., 2016; Lin et al., 2017), during training, and at test time, the shorter edge length is kept at 800. We use a batch size of 16 and train the model for 140 epochs. We use SGD with Nesterov Momentum (Sutskever et al., 2013) as the optimizer with a weight decay of $1e^{-3}$. For the PASCAL RAW dataset, we use the same train/test split as proposed by the original paper. For the Nikon, Sony, and Zurich datasets, we do an 80-20 train/test split of the entire dataset. For RAOD, we use the publicly available validation set as the test set. All dataset splits and statistics are provided in Fig. 16.

## 5.1 Benchmarking Performance on RAWDET-7

Fig. 2 reports the performance of various object detection networks in the proposed RAWDET-7 benchmark, for different quantization levels. Here we observe that naïvely reducing the bit depth through linear scaling and quantization to a low bit depth degrades the quality of fine-grained details that are critical for tasks like object detection. In comparison, the non-linear scaling and quantization methods perform significantly better at all the bit depths. Optimizing only a single additional parameter $\gamma$ (for $\gamma$ -scaling) jointly with the downstream network during training, we observe clear improvements over linearly scaled and quantized inputs, and competitive performance relative to sRGB in several settings. This shows that even a lightweight, task-aware mapping of the input for quantization can lead to meaningful improvements over naïve quantization pipelines. Logarithmic scaling and quantization, albeit fixed, perform on par with or better than learned $\gamma$ scaling. Among the different target bit depths considered, we observe that 4-bit linear scaling and quantization perform worse across all architectures. Improvements are observed by applying a logarithm or the learned $\gamma$ scaling before the quantization. The improvements are on par with the sRGB setting, indicating that simple pre-quantization mappings such as log or learned $\gamma$ can substantially reduce the gap to sRGB-based pipelines in this benchmark, and work towards alleviating the need for an ISP. We also learn separate $\gamma$ during training for day and night images as well as separate $\gamma$ for different sensors indicated by 2-$\gamma$ and 5-$\gamma$, respectively in Fig. 2. We observe a marginal difference in performance while training with multiple $\gamma$ values, while combining logarithmic scaling with learned $\gamma$ scaling provides some gain in detection performance when compared to solely logarithmic scaling for networks trained from scratch.

## 5.2 Benchmarking Large VLMs (LVLMs)

We also benchmark RAWDET-7 on MM-Grounding-DINO for multiple RAW quantization levels and sRGB baselines under various signal scaling techniques. Unfortunately, training or fine-tuning the VLM is computationally infeasible given our limited resources. Thus, we keep the VLM frozen throughout. Linear, log,

Table 2: Experiments with the large VLM, MM-Grounding-DINO (Zhao et al., 2024), with a Swin-T (Liu et al., 2021) backbone. For $\gamma$-scaling and Log + $\gamma$-scaling (including sRGB), we finetune only the $\gamma$ parameter while keeping the VLM frozen; all other methods are evaluated zero-shot. sRGB + $\gamma$ achieves the best overall performance. At matched 8-bit depth and Log + $\gamma$-scaling, RAW performs on par with sRGB without requiring any sRGB-like ISP processing.

| Quantization | Method | mAP | AP50 | AP75 | AP95 |
|---|---|---|---|---|---|
| RGB 8-bit | sRGB | 0.188 | 0.215 | 0.204 | 0.088 |
| | $\gamma$-SCALING | 0.225 | 0.252 | 0.243 | 0.117 |
| | Log + $\gamma$-SCALING | 0.204 | 0.230 | 0.222 | 0.099 |
| 4-bit RAW | Linear | 0.047 | 0.060 | 0.053 | 0.008 |
| | Log | 0.114 | 0.138 | 0.127 | 0.034 |
| | $\gamma$-SCALING | 0.104 | 0.125 | 0.115 | 0.035 |
| | Log + $\gamma$-SCALING | 0.135 | 0.161 | 0.149 | 0.048 |
| 6-bit RAW | Linear | 0.066 | 0.081 | 0.074 | 0.014 |
| | Log | 0.149 | 0.182 | 0.164 | 0.042 |
| | $\gamma$-SCALING | 0.177 | 0.206 | 0.193 | 0.072 |
| | Log + $\gamma$-SCALING | 0.198 | 0.230 | 0.217 | 0.085 |
| 8-bit RAW | Linear | 0.074 | 0.091 | 0.083 | 0.017 |
| | Log | 0.151 | 0.186 | 0.167 | 0.041 |
| | $\gamma$-SCALING | 0.177 | 0.207 | 0.193 | 0.070 |
| | Log + $\gamma$-SCALING | 0.202 | 0.233 | 0.220 | 0.090 |
| 12-bit RAW | Linear | 0.075 | 0.092 | 0.084 | 0.017 |
| | Log | 0.152 | 0.187 | 0.168 | 0.041 |
| | $\gamma$-SCALING | 0.167 | 0.196 | 0.183 | 0.066 |
| | Log + $\gamma$-SCALING | 0.201 | 0.232 | 0.219 | 0.089 |

| Linear | Log | $\gamma$-SCALING | Log + $\gamma$-SCALING | sRGB | Ground Truth |
|---|---|---|---|---|---|

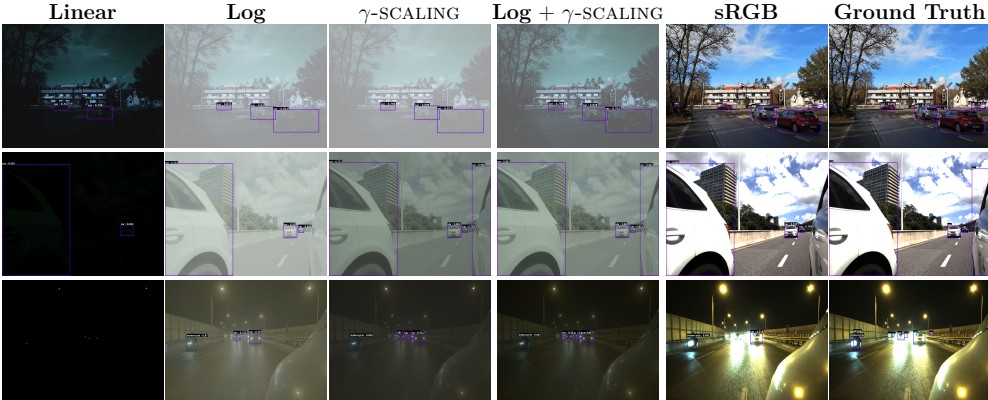

Figure 3: Visualizing predictions made by MM-Grounding-DINO on RAWDET-7 at 6-bit quantization, compared to predictions on sRGB images and the Ground Truth predictions. Here for methods $\gamma$ and Log + $\gamma$, we finetune only the $\gamma$ parameter while keeping the VLM frozen; for the other methods, we evaluate zero-shot. We randomly sample three distinct scenarios to showcase RAWDET-7's versatility.

and plain sRGB are evaluated zero-shot, while for $\gamma$-scaling and log+$\gamma$-scaling we optimize only the scalar $\gamma$ parameter. To make the RAW-vs-sRGB comparison symmetric, we additionally evaluate sRGB+$\gamma$ and sRGB+log+$\gamma$ under the same one-parameter calibration protocol. Thus, for these, we keep the VLM frozen and only train the $\gamma$ value with lr 1e-5 and a maximum training budget of 50 epochs. Results in Tab. 2 show that sRGB+$\gamma$ obtains the strongest overall performance, which is expected for a frozen foundation detector pretrained on sRGB-like inputs, where a learned scalar can further align the input statistics with the model's native modality. Importantly, at the same 8-bit depth and with the same log+$\gamma$ calibration, RAW performs essentially on par with sRGB, showing that suitably processed RAW remains fully competitive under symmetric lightweight adaptation. We also observe that log+$\gamma$ improves all RAW variants over their corresponding $\gamma$-only variants, while the same preprocessing reduces sRGB performance relative to sRGB+$\gamma$, indicating that the most effective lightweight scaling is modality-dependent.

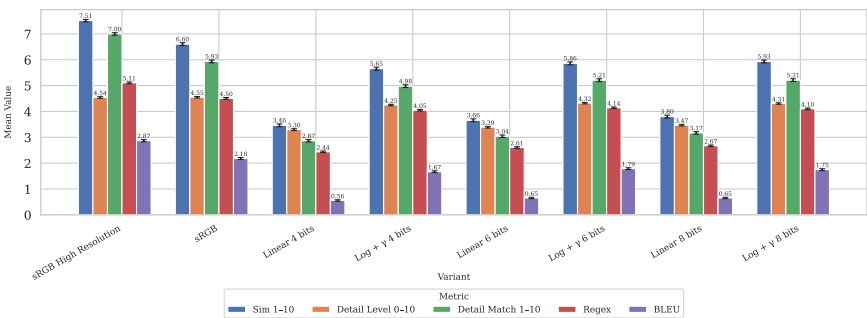

Figure 4: Object Description quality across different image variants. Descriptions are generated from linear and log+γ variants of quantized images using Gemini, with reference captions obtained from higher-resolution sRGB images. Here, sRGB refers to the sRGB images downsampled 2× to match the resolution of the RAW images after extracting R, G, and B channels. We multiply BleU and Regex scores by 10 to align them with the other metrics. Black lines indicate standard errors. The results show that captions generated from processed RAW images (log+γ) achieve competitive quality, closely matching the reference captions.

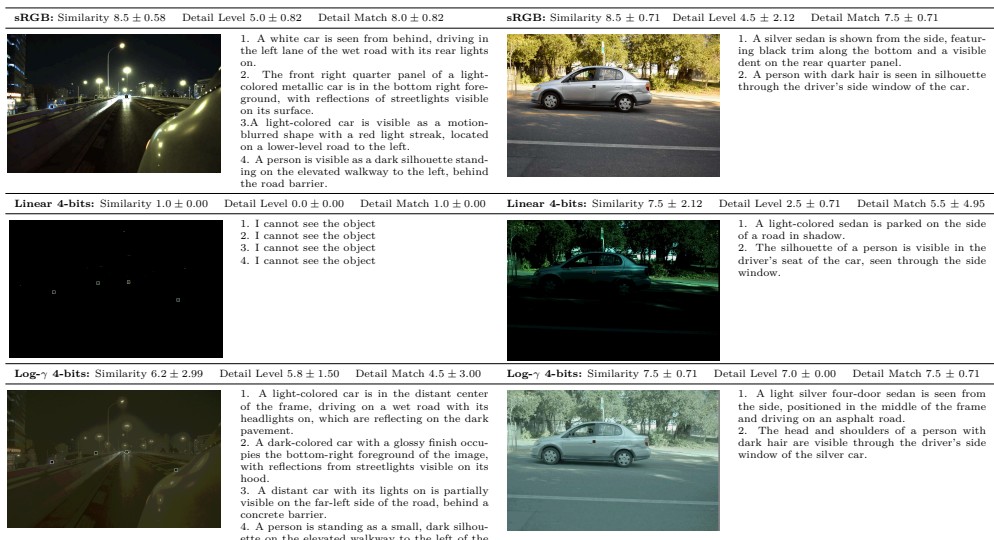

Figure 5: Caption quality for different image representations, including quantitative scores. Captions generated from Log-γ images provide more detail-level explanations than Linear 4-bit quantized images.

## 5.3 Object Description

Next, we study object-description quality across different RAW preprocessing pipelines. Since current closed VLMs cannot be fine-tuned end-to-end on RAW data, we adopt a zero-shot setting and reuse the γ values that showed strong robustness in the object detection experiments across architectures and bit depths. We randomly sample 500 images from the RAWDET-7 validation set and treat high-resolution sRGB as the reference modality, not absolute semantic ground truth. Each image is annotated with set-of-marks and provided twice to Gemini-2.5-Pro (Comanici et al., 2025) to assess the stability of sRGB object descriptions.

Unlike object detection annotations, which provide precise localization but only coarse class labels, object descriptions allow us to evaluate agreement with high-resolution sRGB-Gemini descriptions, i.e., how much VLM-accessible object-level information remains recoverable after RAW preprocessing. They therefore provide a reference-based, not absolute, consistent reference for controlled, fine-grained comparisons across preprocessing methods and image variants.

From the corresponding Bayer-patterned RAW measurements, we extract R, G, and B channels and construct quantized RAW variants at 4, 6, and 8 bits using two mappings: linear and log + γ, where the latter combines

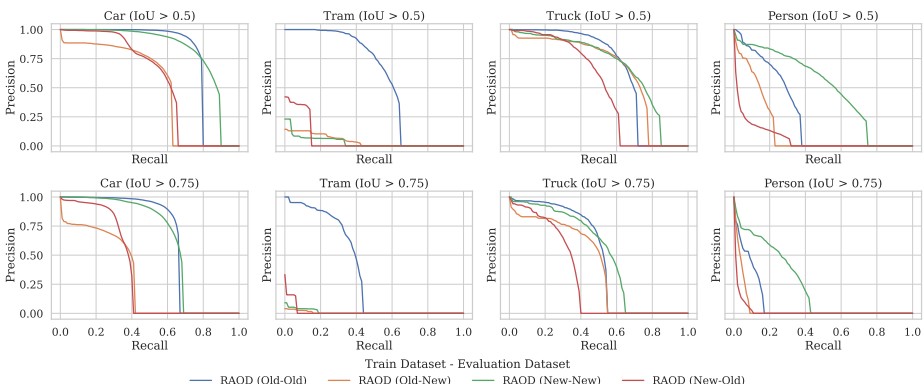

Figure 6: Precision Recall Curves for RAOD. "Old" refers to the annotations as proposed in the original dataset, whereas "New" means the annotations in RAWDET-7. All the models are trained on FasterRCNN, for 8-bit quantization and jointly learnt with 1 gamma. As mentioned in the legend, the keys are "(Train Dataset - Evaluation Dataset)".

logarithmic compression with the $\gamma$ values. We additionally generate a $2\times$ downsampled sRGB image whose effective resolution roughly matches that of the 8-bit RAW variants. All non-reference representations, namely downsampled sRGB, linear RAW, and $\log + \gamma$ RAW, are overlaid with the same set-of-marks and passed to Gemini-2.5-Pro for object descriptions. Caption quality is evaluated against the high-resolution sRGB-Gemini reference using five metrics: BLEU, Regex Match, Similarity, Detail Match, and Detail Level (all 1-10), where the last three are scored by Gemini-2.5-Flash-Lite for consistent qualitative assessment. Formal metric definitions are in Sec. J.3; human- and Claude-based judge checks are in Secs. J.5 and J.6.

The results in Fig. 4 show a clear and consistent pattern. Across all bit depths, linearly quantized RAW images produce captions that are less similar to the sRGB-Gemini reference and substantially worse in detail and detail match, indicating that naive quantization removes image structure used by the captioning model. In contrast, $\log + \gamma$ quantized RAW images achieve much higher agreement with the high-resolution sRGB-Gemini reference and recover more VLM-accessible object-level detail, while the $2\times$ downsampled sRGB variant lies between these extremes. This supports our central claim that careful RAW preprocessing can substantially narrow the gap between what an RGB-pretrained VLM can describe on sRGB and on RAW inputs. At the same time, the remaining gap and the strong sensitivity to preprocessing highlight that poor RAW processing leads to systematically worse descriptions, whereas good RAW-aware mappings are feasible and worth optimizing, as illustrated in Fig. 5. The object-description track in RAWDET-7 makes these trade-offs measurable and enables future work to benchmark RAW preprocessing pipelines not only against sRGB, but also across multiple bit-depth-accurate RAW variants. Looking ahead, optimizing $\gamma$ specifically for object description may further improve performance. We also report metrics after excluding refusals such as "I cannot see the object" in Figure 18 in Appendix Section J.

### 5.4 Discussion: Added Value by the Contributed Annotations and Descriptions

Fig. 6 shows the precision-recall curves for the RAOD dataset in four different settings. "Old-Old" means that the model was trained with original annotations and tested with them as well, whereas "Old-New" indicates a different test setting, testing with newly proposed annotations, and likewise for "New-New" and "New-Old". Since the annotations in both cases are slightly different, we only compute the precision-recall curves for 4 mutually common annotations i.e. *'Car'*, *'Tram'*, *'Truck'*, and *'Person'*. We observe that the precision when tested with new annotations is lower for the same recall value. This indicates that the difficulty level of these annotations is higher compared to the original dataset. We have added separate plots for day and night images in the appendix (Fig. 14 and Fig. 15). Fig. 7 shows that training on the proposed RAWDET-7 dataset improves performance across individual datasets, compared to training the model individually with every subset. This indicates that combining multi-bit depth raw images during training helps the model generalize well on every subset of data, which further proves the usefulness of RAWDET-7. We provide results for PAA in appendix Fig. 12

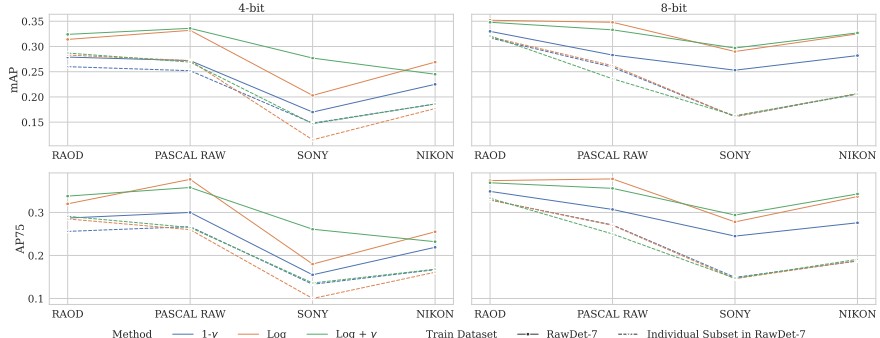

Figure 7: Results for training on combined RAWDET-7 when evaluated on each subset, vs. training only on a subset and evaluated the subset's test set. Combining improves results on all subsets. The model used is Faster RCNN.

While object detection annotations provide precise localization, they remain limited to coarse, class-level characterization of scene content. To assess RAW preprocessing, we compare the extent to which the focus on object-level semantic, spatial, and contextual details is retained in Gemini descriptions relative to sRGB. These annotated descriptions provide a consistent reference-based evaluation, enabling controlled, fine-grained comparisons across preprocessing methods and RAW variants.

## 6 Conclusion

RAWDET-7 addresses a key bottleneck in RAW-domain vision by providing a large-scale, standardized, and human-validated benchmark for object detection and object description on quantized RAW images. By consolidating and re-annotating four existing datasets, we create a diverse benchmark spanning varied sensors, lighting conditions, environments, and bit-depth regimes, while substantially improving annotation quality, category coverage, and the labeling of small and partially occluded instances. Beyond detection, the object-description track extends evaluation from class-level recognition to fine-grained semantic, spatial, and contextual understanding, providing a new way to study how much VLM-accessible object-level information remains aligned with high-resolution sRGB-Gemini references after RAW processing.

RAWDET-7 further enables controlled benchmarking under realistic low-bit sensing constraints. Across standard detectors and a large grounding model, our experiments show that suitable RAW-aware input mappings make heavily quantized RAW highly competitive with standard sRGB pipelines, while under our human-validated reference-based description protocol, achieving substantially higher agreement with high-resolution sRGB-Gemini reference descriptions than naïve quantization. Supported by human studies validating both the annotation and description pipelines, we believe RAWDET-7 can serve as a strong foundation for future research in low-bit sensing, quantization-aware perception, and sensor-to-task co-design.

## 7 Limitations

RAWDET-7 improves coverage of sensor settings by building on diverse prior datasets, but its coverage of the broader RAW sensor landscape remains limited. The dataset is also imbalanced across sensors and bit depths, with most images at 24-bit. Even if the practical impact is unclear, a more balanced sensor representation is desirable. In principle, additional RAW datasets could be merged into RAWDET-7 and annotated using our pipeline, but expanding annotations is costly because high-quality annotation tools (including the one we use) are paid. Additionally, as the ground-truth annotations were generated by a foundation model from standard sRGB views, they may miss objects visible only under RAW's high-dynamic-range (HDR) conditions, potentially penalizing valid RAW detections as false positives.

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

# RAWDET-7: A Multi-Scenario Benchmark for Object Detection and Description on Quantized RAW Images

## Supplementary Material

## A Experimental Details

Following, we provide more details regarding the object detection experimental setup: training details for the architecture and details on using the foundation model for generating the new annotations for RAWDET-7.

### A.1 Object Detection Training Details

Following previous works, for training traditional object detection models, we follow multi-scale training where the shorter image side is scaled to one of the sides randomly selected from a set of sides: [480, 512, 544, 576, 608, 640, 672, 704, 736, 768, 800] using nearest neighbor interpolation and the longer side is scaled

to maintain the aspect ratio. We apply a warm-up for the first 1000 iterations, linearly increasing the learning rate (lr) from $1e^{-3}$ to $2.5e^{-3}$ followed by a multi-step learning rate scheduler.

### A.2 Details of Foundation Model used for Annotations

We used the paid API version at: `https://deepdataspace.com/request_api`. We use their model "DetectionModel.GDino1_5_Pro" for generating the bounding boxes and classes such that we only annotate objects with 20% or less overlap with another object of the same class (following recommended settings). The text prompt we used for generating the annotations was "car . truck . tram . person . bicycle . motorcycle . bus .". Each query to the API costs ¥0.1, thus, including the debugging and ablations over prompts and classes, the total cost was ¥3402.80. Finally, after obtaining the annotations, we observed that the model was often hallucinating low-confidence objects. Thus, we filter out these hallucinated objects, after carefully observing multiple annotations and confidences, by using a confidence score threshold of 0.8, i.e. only annotations with confidence scores of 0.8 and above were retained, and the remaining were filtered out.

## B   User Study Comparing Our New Object Detection Annotations to Old

We did a user survey for our new data annotations and received 53 responses. For the survey, we collected 100 random pairs of annotations such that for each image chosen at random, we had the annotation from RAWDET-7 and the annotation provided by the original dataset. We split the 100 random pairs of annotations into 4 sets of 25 questions each. Each question shuffled the order of option A and option B, with annotation from RAWDET-7 for that image being either option A or option B, and the other option being the annotation from the original dataset.

Given 25 questions to each user, and since 53 users responded, we had exactly 1325 instances of comparisons between the RAWDET-7 annotations and the original annotations answered. From these 1325 instances, users preferred the RAWDET-7 annotations in 1002 instances. That is, **users preferred** RAWDET-7 **annotations for 75.62% of all the instances**. This clearly demonstrates the improvement of the annotations provided by RAWDET-7. For ease of access to the comparisons, in Section E, we extended the comparison from Figure 1 and show a few examples comparing the annotations from RAWDET-7 and the original annotations.

**Limitations Of The User Study.**     Post the study, a few users informed that they had to zoom in quite a bit to confirm that one annotation is better than the other. We suspect that the annotations that required zooming in were the annotations from RAWDET-7 since it provides very fine-grained detections as well. Users reported that, in a few instances, those who are not vigilant may not choose the RAWDET-7 annotations, and thus the final scores might underrepresent the benefit of the annotations from RAWDET-7.

## C   Annotation Quality Validation

To assess absolute annotation quality, we manually inspected 80 images per source dataset, stratified across lighting conditions, yielding 480 images total. A single expert annotator recorded TP, FP, and FN for each class using IoU $\geq 0.5$ as the match criterion, assessed by visual inspection. In the vast majority of TP cases, predicted bounding boxes exhibited clear, correct overlap with the target object. Per-class precision and recall, broken down by class and lighting condition, are reported in the Fig. 9. Across the 480 manually inspected images, the Grounded-DINO detector at a confidence threshold of 0.8 demonstrated consistently high annotation quality for the predominant object categories. Cars and persons, the two most frequent classes, achieved precision and recall above 0.95 in nearly all dataset-lighting combinations, confirming robust detection under both day and night conditions. Bicycles and motorcycles showed slightly more variability, with precision occasionally dropping to 0.75–0.88, primarily due to a small number of false positives rather than systematic misdetection. Trucks exhibited the greatest precision variance across datasets (0.556–1.000), attributable to low instance counts in several subsets amplifying the effect of individual false positives. The tram class was the notable outlier: with zero true positives across four of the six datasets, it reflects genuine scarcity in the data rather than detector failure. Excluding tram, and bus in datasets where it was absent, the

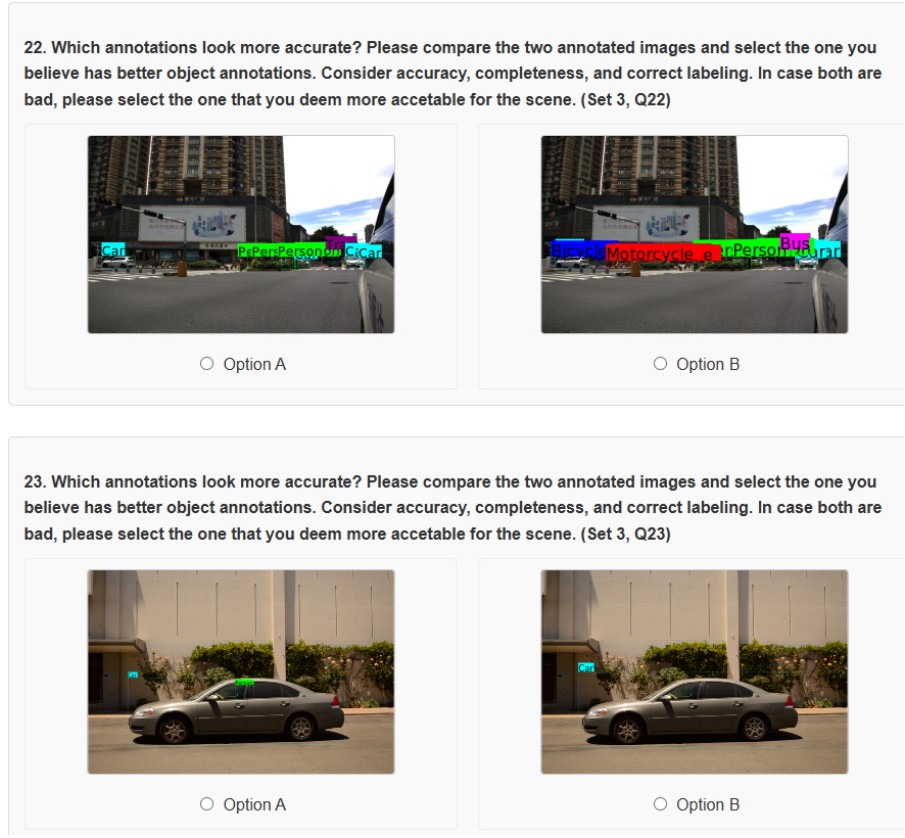

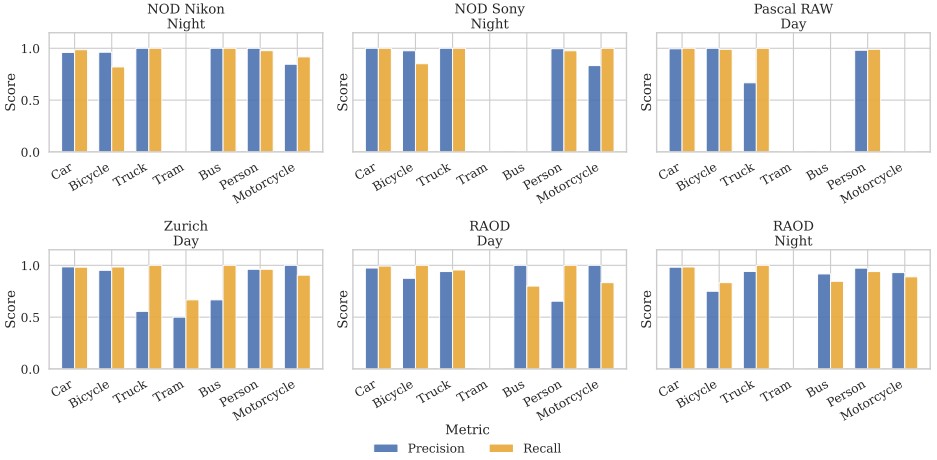

Figure 8: Examples from the annotation-quality user study, where participants inspected bounding boxes and compared newly generated annotations against older versions to assess improvements in quality.

Figure 9: Distribution of per-class precision and recall from manual inspection of 80 images per source dataset (480 images total), stratified across lighting conditions. Each panel corresponds to one dataset–lighting combination. Classes with zero instances in a given subset (e.g. tram in most datasets) are shown as zero and reflect genuine data scarcity rather than detector failure.

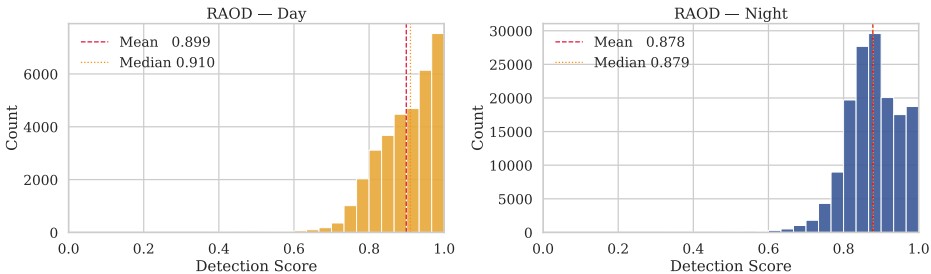

Figure 10: Distribution of Grounded-DINO 1.5 confidence scores for daytime and night-time images in the RAOD dataset. The distributions show substantial overlap, with mean scores of 0.899 and 0.878 respectively.

Table 3: Characteristics of Annotated Objects in RAWDET-7.

|  | Bicycle | Bus | Car | Motorcycle | Person | Tram | Truck |
|---|---|---|---|---|---|---|---|
| Avg. Obj Size (H × W) | 477 × 431 | 224 × 301 | 250 × 257 | 231 × 195 | 508 × 213 | 177 × 307 | 231 × 257 |
| Avg. Obj Count/Image | 1.95 | 1.28 | 7.63 | 2.12 | 3.62 | 1.12 | 1.53 |
| Avg. Aspect Ratio | 0.939 | 1.486 | 1.350 | 0.888 | 0.528 | 2.192 | 1.197 |

macro-averaged precision and recall across all classes and conditions consistently exceeded 0.90, supporting the use of these annotations as a reliable ground truth for benchmarking purposes. These results confirm the reliability of 0.8-confidence Grounded-DINO outputs as annotation sources for the benchmark. One limitation, though, is that the post hoc human audits of model-assisted annotations may be subject to confirmation bias, as the human can also overlook the objects missed by the model.

# D Confidence Score Analysis by Lighting Condition

To assess whether the fixed confidence threshold of 0.8 introduces a systematic annotation bias between daytime and night-time images, we analysed the distribution of Grounded-DINO 1.5 confidence scores across RAOD - day and night datasets, broken down by lighting condition. Figure 10 shows the distribution of confidence scores for daytime and night-time images in RAOD. The mean confidence score was 0.899 for daytime images and 0.878 for night-time images, a difference of less than 0.02. The distributions overlap substantially across both conditions, suggesting that the foundation model does not exhibit a strong systematic confidence drop on night-time or low-light images in our data.

# E Visualizing And Comparing Annotations From RAWDET-7

In Figure 11, we observe that RAWDET-7 covers many small objects in the images that were originally missing from the annotations, and correctly labels many misclassified objects in the original annotations.

# F Additional Dataset Statistics

In Table 3, we list different object characteristics in the proposed RAWDET-7dataset. Average object size is computed by averaging the height and width of bounding boxes across all images in the train and validation sets for a particular category. We observe that the bicycle category has the largest object size, and the motorcycle, truck, and tram have the smallest. The category "car" has the highest average object count per image, which is intuitive due to the outdoor nature of the proposed dataset. Average aspect ratio is computed by calculating the aspect ratio (width/height) of every object in a category and averaging over the entire dataset.

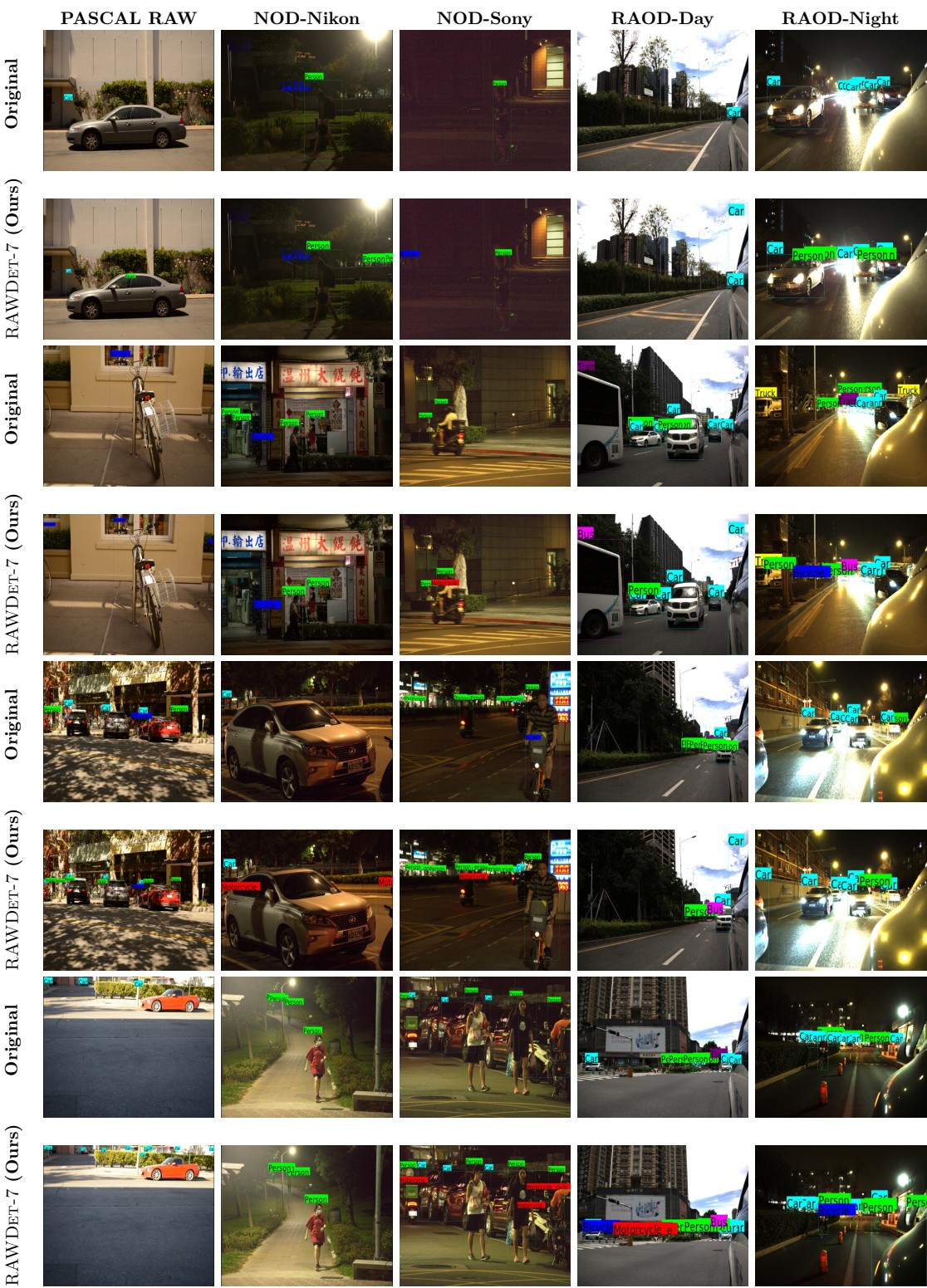

Figure 11: Comparing ground truth annotations provided in the original datasets and the new ones proposed in RAWDET-7 as used in the user study. Please note that the visualizations here are heavily compressed due to size limitations.

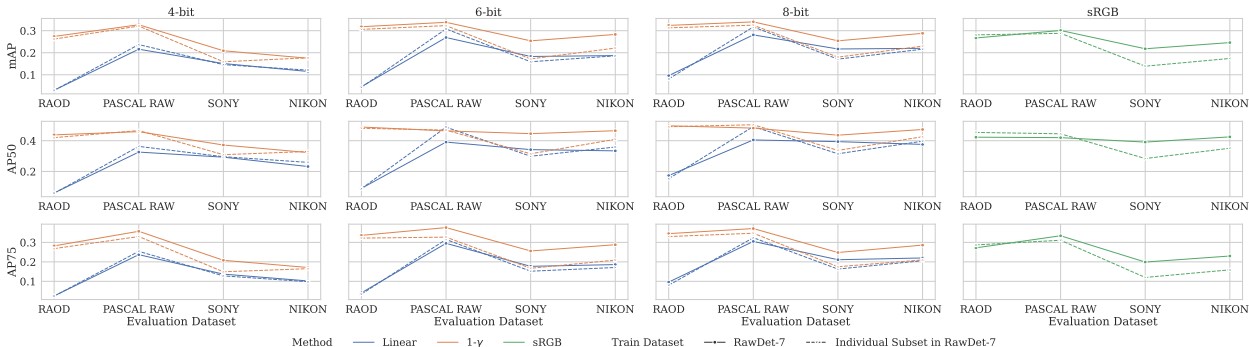

Figure 12: Results with PAA for training on combined RAWDET-7 when evaluated on each subset, vs. training only on a subset and evaluated on the subset's test set. Combining improves results on the majority of the subsets in the case of gamma scaling and sRGB. These comparisons include all metrics (mAP, AP50, AP75) and cover 4, 6, and 8-bit depth data as well as sRGB.

## G  Additional Benchmarking Details e.g. details on the ISP used for RAW RGB conversion of the images from the RAOD dataset.

Except for the Zurich Dataset, the remaining individual datasets have a Bayer pattern of RGGB, whereas Zurich raw images have an RGBG pattern. We extract red, green, and blue channels from the relevant Bayer pattern and take the mean of the double green channels. For quantization experiments, we use *torch.floor* as a quantization function. In the case of gamma scaling, we use a straight-through estimator to let the gradients pass through the quantization operation. The RGB images of all the individual datasets are publicly provided along with the raw images, except for the RAOD dataset. So, we generated them by first extracting the Red, Green, and Blue channels from the original raw image, followed by a gray world white balance algorithm. Then, we apply gamma correction, where the value of gamma is chosen by hit-and-run to be 0.09.

## H  Advantage of Consolidated Dataset

We show the advantage of adding all the individual datasets together for training vs. training only with the individual datasets in Figure 12 for PAA architecture. We observe that while training with linear quantization without any scaling results in no performance gain when trained in a combined fashion, gamma scaling and RGB experiments usually benefit from combined training. This is especially visible in the case of the Sony and Nikon Datasets.

In Figure 13, we show results with FRCNN for training on combined RAWDET-7 when evaluated on each subset, vs. training only on a subset and evaluated on the subset's test set. Combining improves results on the majority of the subsets in the case of gamma scaling and sRGB. These comparisons include all metrics (mAP, AP50, AP75) and cover 4, 6, and 8-bit depth data as well as sRGB.

In Figure 6, we show precision-recall curves for the RAOD dataset. "Old" refers to the annotations as proposed in the original dataset, whereas "New" refers to the annotations in RAWDET-7. All the models are Faster R-CNNs, trained for 8-bit quantization and jointly learnt with 1 gamma. As mentioned in the legend, the keys are "(Train Dataset - Evaluation Dataset)". Curves are shown for the classes car, truck, tram, and person. Similar trends are observed when the evaluation is performed separately on daytime and night-time subsets of RAOD, suggesting that the annotation improvements are consistent across lighting conditions. (see Figure 14 and Figure 15)

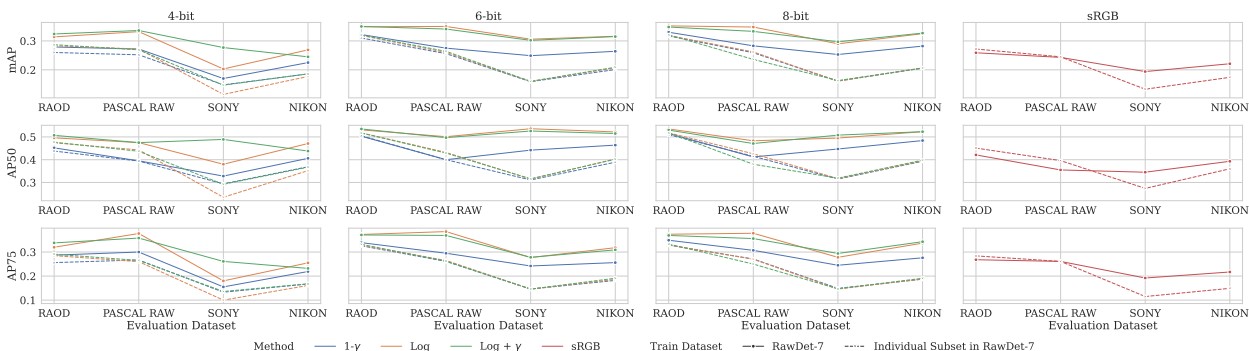

Figure 13: Results with FRCNN for training on combined RAWDET-7 when evaluated on each subset, vs. training only on a subset and evaluated on the subset's test set. Combining improves results on the majority of the subsets in the case of gamma scaling and sRGB. These comparisons include all metrics (mAP, AP50, AP75) and cover 4, 6, and 8-bit depth data as well as sRGB.

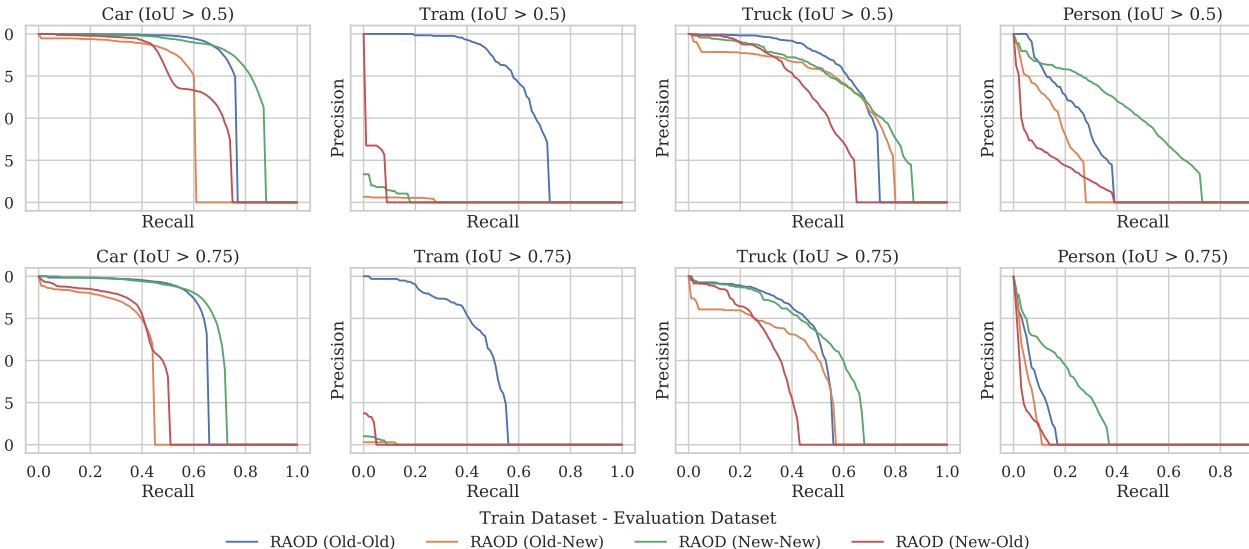

Figure 14: Precision Recall Curves for RAOD-Day images. "Old" refers to the annotations as proposed in the original dataset, whereas "New" means the annotations in RAWDET-7. All the models are trained on Faster R-CNN for 8-bit quantization and jointly learnt with 1 gamma. As mentioned in the legend, the keys are "(Train Dataset - Evaluation Dataset)". Curves are shown for the classes car, truck, tram, and person.

In Table 1, we summarize the statistics on the different datasets that we consolidate into RAWDET-7. The individual datasets have each a particular sensor model and bit depth. The combined dataset, therefore, provides significantly increased data diversity.

In Figure 16, we provide detailed dataset statistics. In particular, we report statistics on instances in the original proposed datasets indicated by 'OLD' and the re-annotated images for those respective datasets indicated by 'NEW' for the classes in RAWDET-7. The proposed RAWDET-7 is a consolidation of RAW input images from the 'OLD' datasets provided with the 'NEW' annotations. For some classes, it appears that 'OLD RAOD' has more instances than the proposed RAWDET-7. Note that RAOD has a large number of objects annotated. However, as shown in Figure 1, the original annotations of the RAOD dataset contain hallucinations for classes like 'Tram', 'Truck', and 'Car'.

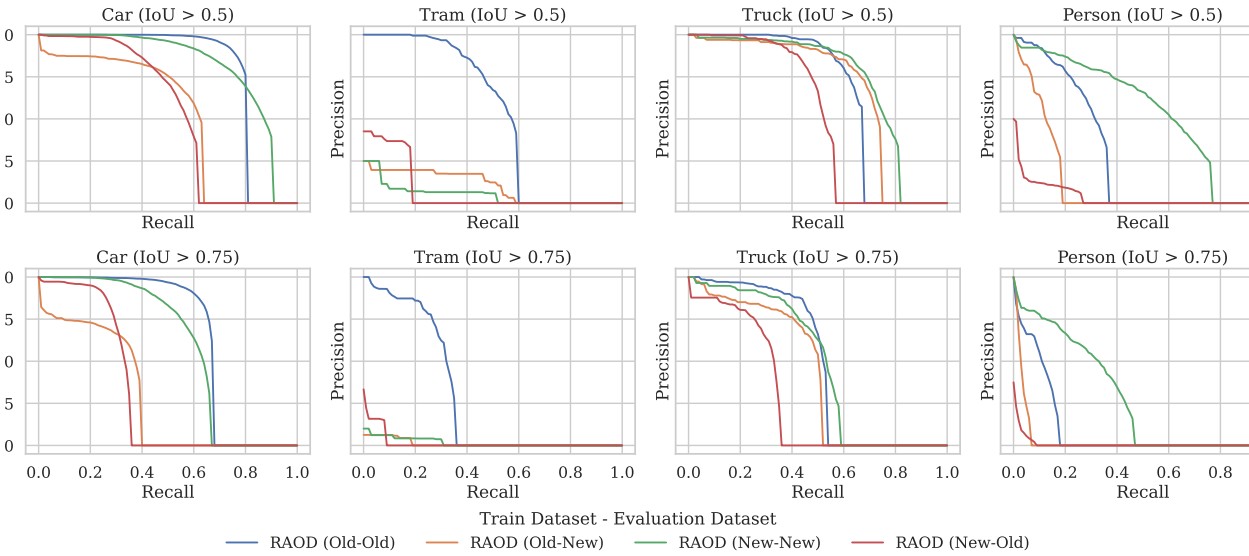

Figure 15: Precision Recall Curves for RAOD-Night images. "Old" refers to the annotations as proposed in the original dataset, whereas "New" means the annotations in RAWDET-7. All the models are trained on Faster R-CNN for 8-bit quantization and jointly learnt with 1 gamma. As mentioned in the legend, the keys are "(Train Dataset - Evaluation Dataset)". Curves are shown for the classes car, truck, tram, and person.

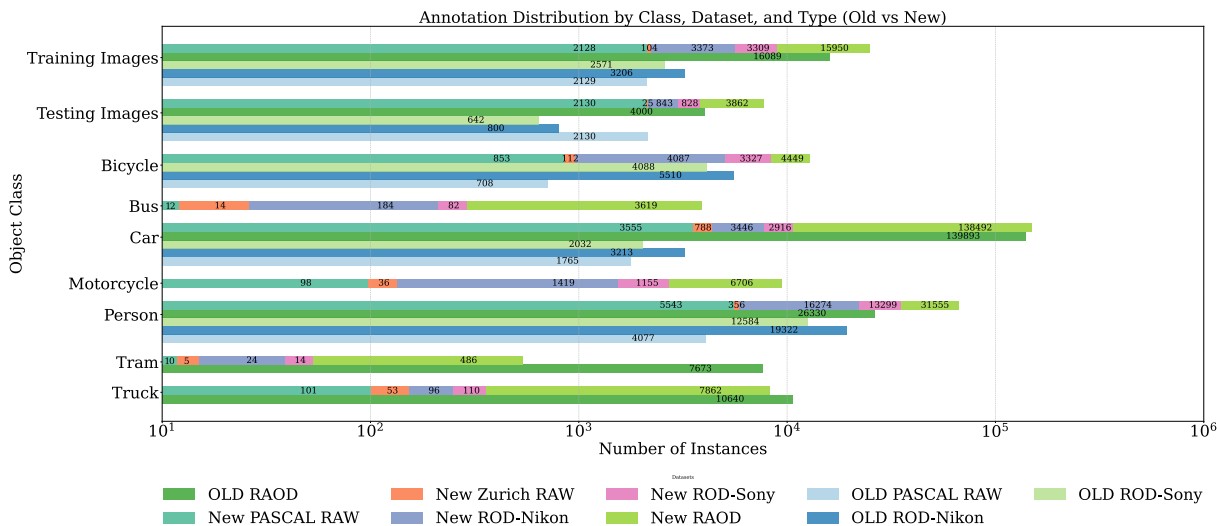

Figure 16: Dataset Statistics: Breakdown of instances in the original proposed datasets indicated by 'OLD' and the re-annotated images for those respective datasets indicated by 'NEW' for the classes in RAWDET-7. The proposed RAWDET-7 is a consolidation of RAW input images from the 'OLD' datasets provided with the 'NEW' annotations. For some classes, it appears that 'OLD RAOD' has more instances than the proposed RAWDET-7. However, as shown in Fig. 1, the original annotations of the RAOD dataset contain hallucinations for classes like 'Tram', 'Truck', and 'Car'.

Table 4: Detection performance (mAP, mAP$_{50}$, and mAP$_{75}$) under different bit-depth and color encoding settings.

| Setting | mAP ↑ | mAP$_{50}$ ↑ | mAP$_{75}$ ↑ |
|---|---|---|---|
| 4-bit RGB | 0.303 | 0.485 | 0.316 |
| 4-bit RGGB | 0.290 | 0.483 | 0.301 |
| 6-bit RGB | 0.351 | 0.545 | 0.371 |
| 6-bit RGGB | 0.340 | 0.545 | 0.357 |
| 8-bit RGB | 0.361 | 0.558 | 0.383 |
| 8-bit RGGB | 0.343 | 0.550 | 0.359 |

The consolidated dataset RAWDET-7 provides cleaned labels for the diverse RAW data samples.

## I   Results with 4-channel input: RGGB

We conduct experiments (see Table 4) by extracting the four Bayer channels (R, G, G, B) from the raw images and modifying Faster R-CNN to accept 4-channel inputs. Both RGGB and RGB experiments were trained jointly with a 1-gamma setting for 140 epochs. We observe that the RGB and RGGB inputs perform on par for various bit depth settings. Please note that this on-par performance is despite the fact that we used a ResNet-50 backbone pretrained on ImageNet, which is optimized for 3-channel RGB images. However, we find this result also intuitive since the information provided to the network in RGB and RGGB is very similar, and only a little extra resolution is provided by RGGB.

## J   Object Descriptions

### J.1   Generating Object Descriptions

For object captioning, we use the `caption prompt used for object-level descriptions` prompt with `Gemini-2.5-Pro`. The model receives the overlaid image where each detected object is marked with a numbered square, together with the list of corresponding class labels. The prompt enforces a strict line-based output format of the form "`<number>:  <caption>`" and requires exactly one caption per numbered object, in order from 1 to N. We call `Gemini-2.5-Pro` on each marked image (twice for the sRGB reference and once for each RAW or RGB-downsampled variant) and then parse the returned text using a regular expression to obtain a dictionary that maps object indices to their captions, which is then used for both analysis and visualization.

### J.1.1 Prompts for Generating Object Descriptions

> **Caption prompt used for object-level descriptions**
>
> ```
> You will be shown a photo with multiple objects with these 7 classes of interest: cars, truck,
>     tram, person, bicycle, motorcycle, bus.
> Each object of interest is marked by a black square containing a coloured number.
>
> Task: For EACH object with a marked number, produce ONE detailed, factual caption that ONLY
>     describes that numbered object.
> Include concrete details when visible: color, material/texture, approximate size class, local
>     position (e.g., top-left),
> distinctive parts, visible text/markings, and immediate context relations.
>
> STRICT OUTPUT:
> Write ONE line per object using this exact pattern: <number>: <caption>
> Acceptable separators after the number are ':', ')', '.', or '-' (e.g., '1: ...' or '1) ...').
> Do NOT include headings, bullets, blank lines, or any extra text before/after the list.
> If a numbered object cannot be seen or identified, write exactly: 'I cannot see the object'.
> The numbering starts at 1 and goes up to N, where N is the total number of marked objects in
>     the image.
> There should be exactly N lines in your output, one for each numbered object.
> If N is the highest number visible, ensure you include captions for all numbers from 1 to N in
>     order.
> We will tell you the object class of each marked object in the following way:
> 1: <CLASS>
> 2: <CLASS>
> ...
> N: <CLASS>
>
> If a numbered mark itself cannot be seen or identified in the image, write exactly: 'I cannot
>     see the mark'.
> DO NOT HALLUCINATE ANY MARKED NUMBERS.
> DO NOT TALK ABOUT THE MARK SQUARES OR NUMBERS ITSELF or the WHOLE IMAGE.
> ```

### J.2 User Study to Validate Quality of the Object Descriptions

We conducted a user study to evaluate the effectiveness of generated captions. We selected 100 images with the highest mean detail levels and then randomly sampled 20 images and their respective captions from these for each user study. We chose the 100 images with the highest mean detail level, since usually it is the longer output text where generative models hallucinate the most, as they often try to make up details. We wanted the user study to validate that this is not the case in our proposed object descriptions. Participants were shown each image along with its captions and asked to rate the quality of the captions on how well they described the marked objects in each image on a scale from 0 to 5. Here, 0 meant that the object description does not describe the object at all, and 5 meant that the object description describes the respective marked object reasonably well, leaving out almost no describable visual cues.

A total of 63 users participated in this study, so a total of $63 \times 20$, that is, 1260 samples were collected for the quality of the captions. We received an average rating of $\frac{4.07 \pm 0.48}{5}$, that is $\approx 82\%$ score, which demonstrates the good quality of the object descriptions generated.

### J.3 Metrics for Evaluating Object Descriptions

- **BLEU**: Measures how much the generated text overlaps in wording with a reference.

- **Regex Match**: Checks if the output matches a specific pattern or format.

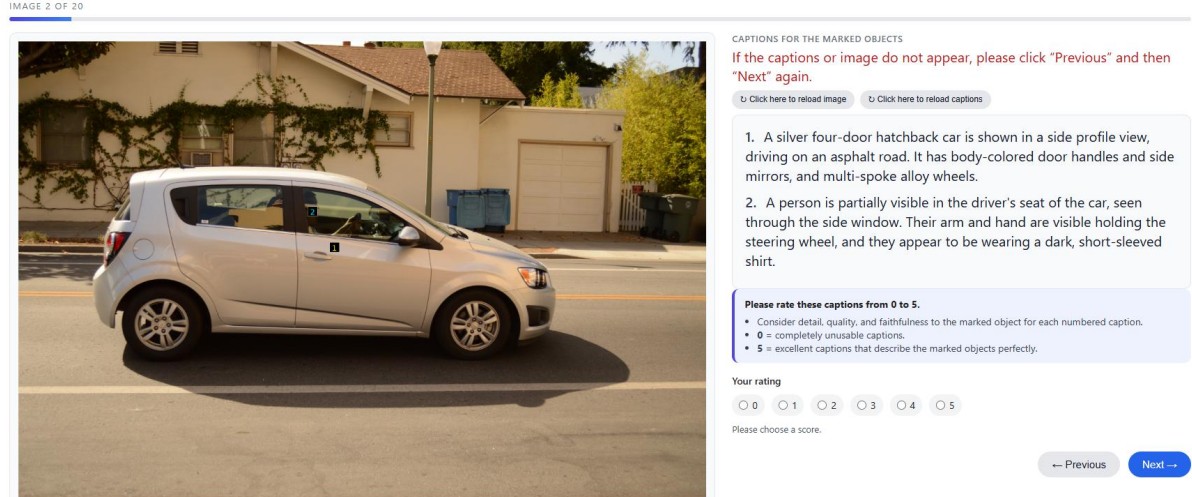

Figure 17: Example from the user study used to assess caption quality. Participants were shown images with designated markers and asked to rate the quality of the accompanying captions, where 0 is completely unusable and 5 is fully descriptive, clear, and highly informative.

- **Similarity (1–10)**: Rates how semantically similar the meaning of two texts are, irrespective of the specific details.

- **Detail Level (0–10)**: Rates how much detail or specificity the output contains.

- **Detail Match (1–10)**: Rates how well specific details in the output match the reference.

**Similarity.** To quantify semantic similarity between two captions that describe the same object, we use the `Similarity scoring prompt` with `Gemini-2.5-Flash-Lite`. This prompt presents Caption A and Caption B and instructs the model to ignore surface-level differences such as wording (for example, "car" versus "sedan") and colour or brightness variations, and to focus instead on whether both captions refer to the same underlying object with consistent meaning. The model is required to output a single integer in [1, 10] on a separate line, where 1 denotes unrelated or contradictory descriptions and 10 denotes near-identical semantics. We use this score as the *Similarity* metric for each caption pair, where one of the captions is always the first caption generated for the full-scale RGB image.

**Detail Level.** To assess how informative a single caption is, we use the `Detail level prompt` with `Gemini-2.5-Flash-Lite`. The prompt asks the model to rate the richness of grounded detail in one caption only, again returning a single integer score in [0, 10]. The instructions explicitly direct the model to base its judgment on concrete attributes such as materials, positions, parts, readable text, and local relations, while ignoring fluency and style. A score of 0 indicates no grounded detail, whereas 10 corresponds to a very specific, highly descriptive caption. We treat this as the *Detail Level* metric and compute it for each caption that we evaluate.

**Detail Match.** Finally, we measure the consistency of fine-grained information between two captions using the `Detail match prompt`, again with `Gemini-2.5-Flash-Lite`. Here, Caption A (first full-scale sRGB caption) acts as the reference and Caption B as the candidate, and the model is instructed to assign a score in [1, 10] based on the overlap of concrete, verifiable details such as materials, positions, parts, readable text and relations, while explicitly disregarding differences in colours or brightness due to varying image processing. Low scores correspond to almost no shared details, whereas scores near 10 indicate high

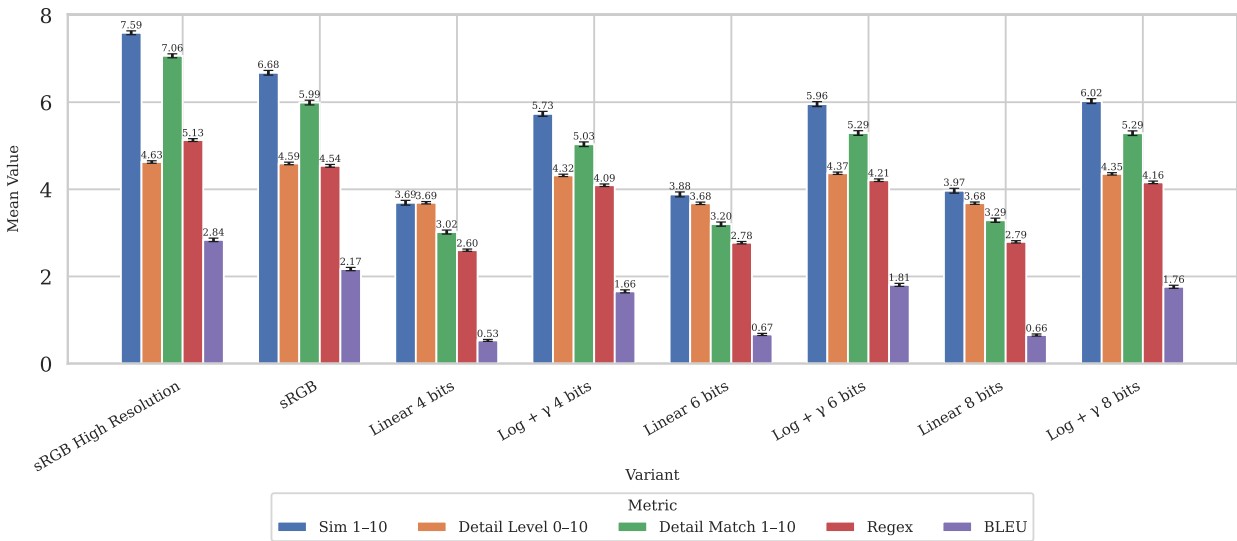

Figure 18: Same as Figure 5, descriptions here are generated from linear and log+$\gamma$ variants of quantized images using Gemini, with reference captions obtained from higher-resolution sRGB images. Here, Object Description quality across different image variants is reported **after excluding the refusals**. sRGB refers to the sRGB images downsampled 2× to match the resolution of the RAW images after extracting R, G, and B channels. We multiply BLEU and Regex scores by 10 to align them with the other metrics. Black lines indicate standard errors. The results show that captions generated from processed RAW images (log+$\gamma$) achieve competitive quality, closely matching the reference captions.

overlap in grounded content. This yields the *Detail Match* metric, which we aggregate over objects and variants to analyze how well different pipelines preserve detailed object-level information.

### J.4  Analysis of VLM Refusals in RAW Object Description

We re-analyse the object-description results separating refusals from scored responses. Table 5 reports refusal rates per preprocessing variant. Linear-RAW exhibits substantially higher refusal rates across all bit depths: 10.58% at 4-bit, 7.72% at 6-bit, and 5.56% at 8-bit. In contrast, all non-linear variants (log, gamma, log+gamma, RGB-downsampled, sRGB) consistently fall below 1.6%, closely matching the sRGB baseline of approximately 1.8%. Figures 4 and 18 show metric scores with and without refusals included, respectively. Removing refusals improves scores for linear-RAW modestly, for example, Detail Level rises from 3.30 to 3.69 at 4-bit and Detail Match from 2.87 to 3.02, but these scores remain substantially below the corresponding log+gamma variants (4.32 and 5.03 at 4-bit). This confirms that the performance gap between linear-RAW and log+gamma stems from two compounding sources: a higher rate of outright refusals, and genuinely lower description quality in the non-refusal responses. The log+gamma transformation addresses both failure modes simultaneously.

Table 5: Refusal rates per preprocessing variant. All variants are evaluated on 2553 images.

| Variant | Bit Depth | Refusals | Rate (%) |
|---|---|---|---|
| sRGB (run 1) | – | 48 | 1.88 |
| sRGB (run 2) | – | 45 | 1.76 |
| RGB-Downsampled | 4 | 37 | 1.45 |
| RGB-Downsampled | 6 | 36 | 1.41 |
| RGB-Downsampled | 8 | 29 | 1.14 |
| Linear-RAW | 4 | 270 | 10.58 |
| Linear-RAW | 6 | 197 | 7.72 |
| Linear-RAW | 8 | 142 | 5.56 |
| Log | 4 | 40 | 1.57 |
| Log | 6 | 26 | 1.02 |
| Log | 8 | 21 | 0.82 |
| Gamma | 4 | 40 | 1.57 |
| Gamma | 6 | 18 | 0.71 |
| Gamma | 8 | 33 | 1.29 |
| Log+Gamma | 4 | 32 | 1.25 |
| Log+Gamma | 6 | 17 | 0.67 |
| Log+Gamma | 8 | 17 | 0.67 |

### J.4.1 Prompts for LLM as a Judge

---

**Similarity scoring prompt**

```
You are scoring the semantic similarity between two captions that describe the SAME object.
Check if they are roughly talking about the same object, ignore details.
Disregard similarity in colour, and disregard difference in brightness of the colours;
    disregard the exact word used, for example, car or sedan; focus on overall meaning match.
Return ONLY one integer in [1,10] on a single line with no other text.
Scoring: 1=unrelated/contradictory, 10=near-identical meaning (paraphrases penalize
    contradictions).
Be strict about the overall meaning match.

Caption A:
{A}

Caption B:
{B}
```

---

**Detail level prompt**

```
You are scoring the richness of a SINGLE caption, that is, how long and detailed and
    descriptive it is.
Return ONLY one integer in [0,10] on a single line with no other text.
Score based on concrete grounded attributes (colors, materials, positions, parts, readable text
    , relations).
Be strict about verifiable details only. 10=very rich and specific, 0=no grounded detail.
Do not look for the length of the caption, only the amount of concrete details.
Ignore fluency/style. 0=no grounded detail, 10=very rich and specific.

Caption:
{C}
```

**Detail match prompt**

```
You are scoring the overlap of concrete details between two captions.
Return ONLY one integer in [1,10] on a single line with no other text.
Score high only if both share consistent verifiable details (materials, positions, parts,
    readable text, relations).
Disregard colour differences, and difference in brightness of the colours as the images are
    processed differently, and colour differences are expected, so disregard them.
Ignore fluency/style. 1=no shared details, 10=high detail overlap.
Be strict about concrete detail match.

Caption A:
{A}

Caption B:
{B}
```

### J.5  User Study to Verify LLM as a Judge for Metrics

In order to validate the usage of LLMs as a judge to measure description similarity and level of detail, we also conducted a user study. Figure 19 provides an example from this user study. We gather a random subset of 50 images with the two object descriptions per marked object in the images, gathered from the high-resolution sRGB version of each image, and ask users to provide scores between 0 and 10 for the description similarity, the amount of details provided, and the agreement of the given details.

Each user rates 10 of the 50 randomly sampled images at a time. We collected 9 such user studies, so 90 images and their respective pair of object descriptions for each marked object were rated, that is, a total of 391 caption pairs. We calculate the correlation in the mean similarity, mean detail level, and mean detail match between Gemini-2.5-Flash-Lite as a judge and a human as a judge from each user study. For similarity, we get a correlation of 0.61, for the detail level, we get a correlation of 0.71, and for the detail match, we get a correlation of 0.58. There are all high positive correlations, demonstrating that Gemini-2.5-Flash-Lite can be used as a judge for comparing two object descriptions at a time. Please note that using a better LLM like ChatGPT5 or Gemini-2.5-Pro as a judge might lead to better correlations with humans as a judge; however, this would incur high costs, and thus using Gemini-2.5-Flash-Lite is the most cost-effective way. Using open-source models for LLM as a judge might seem like a reasonable choice; however, as shown by Agnihotri et al. (2025), open-source models like Qwen3, GPT-oss have significantly low correlation with human judgment compared to closed and paid models.

The specific instructions given to the user, for "human as a judge" are:

---

Thank you for taking part in this study. In each trial, you will see a high-resolution image with numbered marks on several objects. For every numbered object, there are two automatically generated captions that are intended to describe the same marked object.

Your task is to rate the relationship between the two captions for each numbered object. For every object number, you will provide one visibility judgment and four scores on a scale from 0 to 10:

- Is the marked object actually visible? (Dependent on the image) Decide whether the object with this number is fully visible in the image, only partially visible, or not visible at all.

- Semantic similarity (Independent of the image) - Do both captions clearly talk about the same object, ignoring fine-grained detail.

- Level of detail of Caption 1 (Independent of the image) - How rich and specific Caption 1 is, in terms of concrete, verifiable details (independent of the image, verifying if the caption itself has obvious conflicts in the text).

- Level of detail of Caption 2 (Independent of the image) - Same as above, now for Caption 2.

- Detail match (Independent of the image) - How well the concrete details in Caption 1 and Caption 2 agree with each other.

Rating scale (0 to 10)

- 0 - No similarity or no grounded details at all.
- 5 - Moderate similarity or a moderate amount of grounded details.
- 10 - Very high similarity or very rich and specific grounded details.

The 0 to 10 scale applies to the four numeric scores. The visibility question instead uses the options Yes, No, and Only partially.

Only for the visibility question (Is the marked object actually visible?), you should look at the image. For all other scores, please ignore the image itself and judge only the two captions for that object. If a caption contains many details that are internally inconsistent or impossible (for example, mentioning both headlights and taillights when only the front of a car can be visible), please give low scores for the relevant detail and detail-match questions.

Please pay attention that each caption is tied to a numbered object in the image. When you score, always think about the object with that number, not the whole image.

---

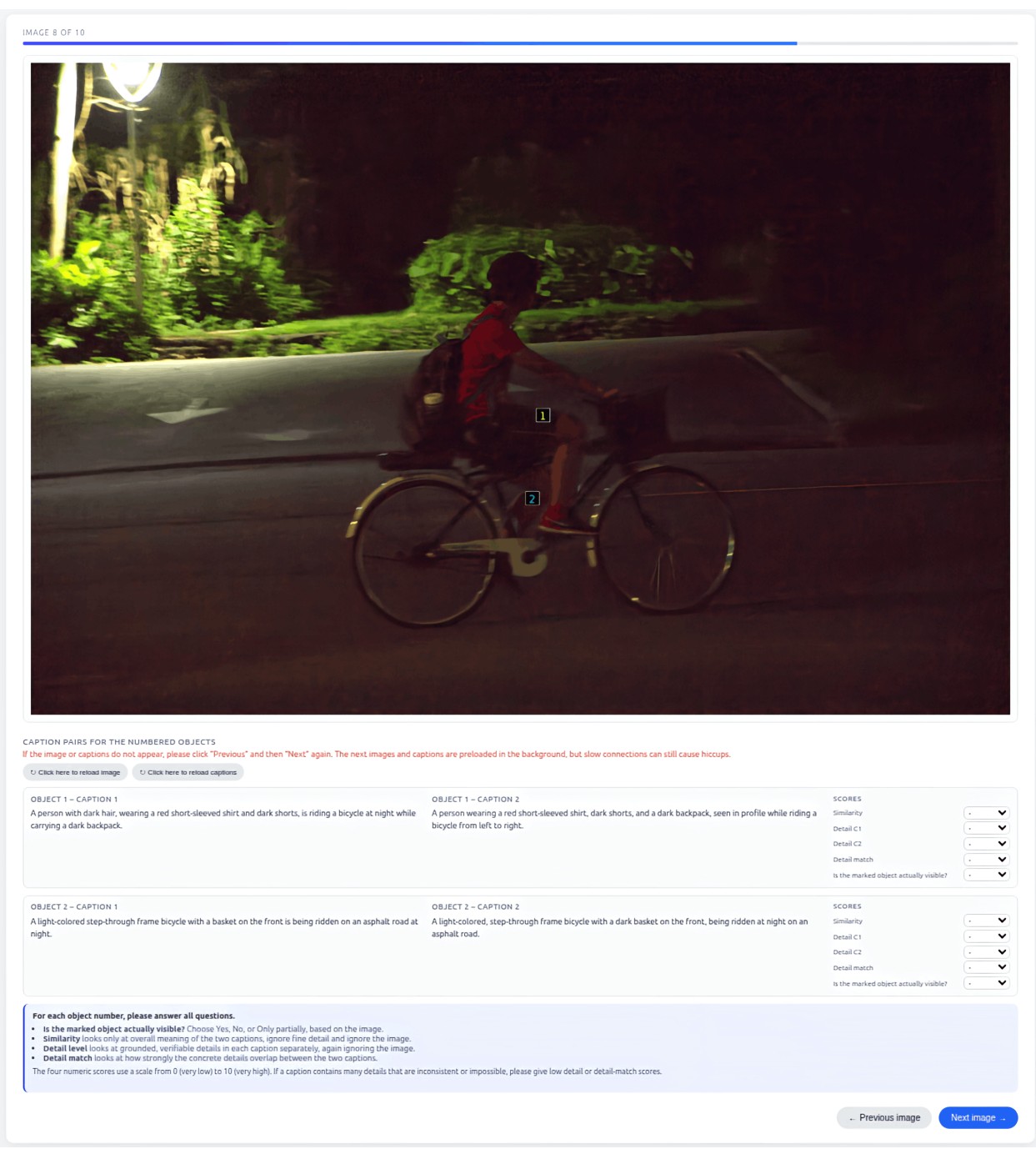

Figure 19: User-study example used to evaluate the performance of an LLM acting as a judge. Participants rated captions based on similarity with one another and richness of details. To assess the captioning model's hallucinations, they also indicated whether the objects described were visible in the images or not.

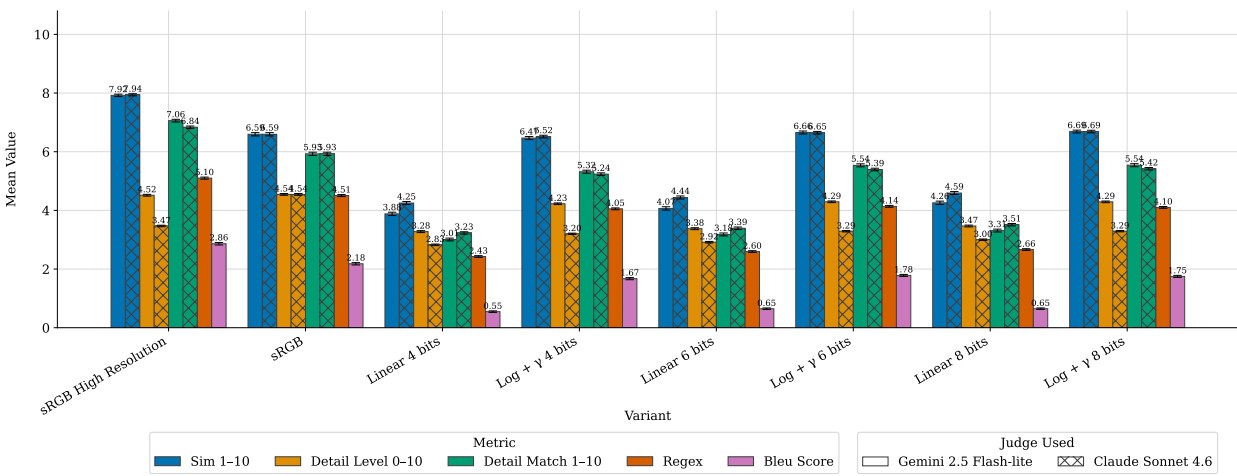

Figure 20: Independent judge validation for the object-description track. We re-score the same caption outputs as in Figure 5 with Claude Sonnet 4.6 Anthropic (2026), while keeping the scoring prompts and aggregation protocol fixed. Claude Sonnet 4.6 agrees strongly with Gemini-2.5-Flash-Lite for Similarity ($\rho = 0.90$), Detail Level ($\rho = 0.70$), and Detail Match ($\rho = 0.91$), and preserves the same relative trends across image representations.

## J.6 Independent Claude Validation of the Description Judge

To address the concern that the object-description track uses Gemini-generated reference descriptions together with a Gemini-family judge, we additionally evaluate the same caption outputs with Claude Sonnet 4.6 Anthropic (2026). We keep the evaluated captions, image variants, scoring prompts, and aggregation protocol fixed, and change only the judge model. As shown in Figure 20, the scores from Claude Sonnet 4.6 are strongly aligned with those from Gemini-2.5-Flash-Lite, with correlations of $\rho = 0.90$ for Similarity, $\rho = 0.70$ for Detail Level, and $\rho = 0.91$ for Detail Match. Importantly, the relative trends across image representations are preserved: descriptions from log+$\gamma$ processed RAW images remain consistently closer to the high-resolution sRGB reference descriptions than descriptions from linearly quantized low-bit RAW images. This independent-judge check does not treat the sRGB reference descriptions as absolute semantic ground truth, but it shows that the reported ordering of RAW processing pipelines is not specific to using a Gemini-family model as judge. Therefore, we use Gemini-2.5-Flash-Lite as the primary judge for the full evaluation due to its lower cost at scale, while the Claude Sonnet 4.6 results provide model-family-independent support for the conclusions of the object-description track.

