# OpenReview forum: "RAWDet-7: A Multi-Scenario Benchmark for Object Detection and Description on Quantized RAW Images"
_TMLR — Decision pending for TMLR_

### Review · Reviewer_wFfa · 2026-04-29

**Summary Of Contributions:**

- **A consolidated and re-annotated RAW detection dataset.** The authors combine 4 existing RAW datasets (PASCALRAW, RAW-NOD-Nikon/Sony, RAOD, Zurich) into a single benchmark of ~25k train / ~7.6k test images, and reannotate them using Grounded-DINO 1.5  with a 0.8 confidence threshold for seven MS-COCO/LVIS-aligned categories. A user study (53 participants, 1325 pairwise comparisons) reports a 75.6% preference for the new annotations over the originals.

- **An object-description track.** For 500 curated images, set-of-mark-style prompting is used with Gemini2.5 on HR sRGB images to produce object-level reference descriptions, which are then compared against descriptions generated from differently processed RAW variants. Quality is measured with BLEU, regex match, and three Gemini2.5 FlashLite metrics.

- **A controlled low-bit benchmark.** The dataset is evaluated under 4 / 6 / 8 bit quantization with 4 input-scaling methods, across Faster R-CNN, RetinaNet, PAA, and MM-Grounding-DINO.

**Audience:**

Yes

**Audience Explanation:**

There is genuine community interest in (1) RAW-domain detection benchmarks, (2) sensor-to-task co-design and quantization-aware perception, and (3) how vision-language models behave on non-standard input regimes. The current state of RAW detection benchmarking is fragmented across several small, idiosyncratically annotated datasets, and a consolidated benchmark with corrected labels and a standardized protocol for low-bit evaluation is a useful resource even if no single technical idea in the paper is new.

**Broader Impact Concerns:**

The paper does not include a Broader Impact statement.

**Claims And Evidence:**

Yes

**Claims Explanation:**

The two core empirical claims of the paper are: (1) RAWDet-7 has higher quality annotations than its constituent datasets, and (2) suitable RAW-aware input mappings make low-bit RAW competitive with, and sometimes superior to, sRGB pipelines. The dataset/scale claims (Table 1, Fig. 13) are supported by simple counting, and the dataset's value for combined training is supported by Fig. 7, Fig. 10, and Fig. 11.

**Requested Changes:**

- Provide absolute annotation quality. The current user study only establishes that the new annotations are preferred over the originals. Please include an absolute-quality assessment, e.g., expert manual correction of a random subset of (say) 200 images per source dataset, reporting precision/recall of the Grounded-DINO outputs against the corrected ground truth, broken down by class and by lighting condition. This is necessary because the entire benchmark rests on the assumption that 0.8-confidence Grounded-DINO outputs on RAW (or sRGB-of-RAW?) are reliable.

- Clarify whether Grounded-DINO was run on RAW or sRGB. Sec. 3.1 says "we re-annotated all datasets using Grounded-DINO 1.5," and elsewhere ("followed by human validation on the corresponding sRGB images") suggests the model was run on sRGB. Please state explicitly which input modality was given to the foundation model and how the resulting boxes were transferred back to the RAW-domain coordinates. If sRGB was used, please state which ISP / which sRGB version was used for each source dataset

- Confidence threshold sensitivity and lighting bias. Foundation models trained on natural sRGB images will systematically have lower confidence on dark/night images (RAOD-Night, NOD), so a fixed 0.8 threshold may under-annotate night scenes. Please report (1) the distribution of confidence scores by source dataset and lighting condition, and (2) recall of new annotations versus the originals broken down by day/night. If there is a systematic skew toward day, this should be discussed as a limitation.

- Address the circularity in the protocol. Please rewrite the framing in Sec. 3.2, 5.3, and the Conclusion to make explicit that the metric measures agreement with sRGB-Gemini outputs, not absolute information preservation. The current language ("preserve substantially richer object-level information than naïve quantization") overclaims. A way to partially break the circularity would be to also include a setting where the ground truth is human-written descriptions on a smaller subset, and report whether the relative ordering of pipelines is the same.

- Clarify the trained-vs-zero-shot comparison in Table 2. The MM-Grounding-DINO results are presented as if they are like-for-like, but γ-scaling and log+γ involve training one scalar parameter while linear/log/sRGB are evaluated zero-shot. Please either (1) add a baseline where a single scalar gain is trained on top of sRGB inputs as well (so the comparison is fair), or (2) explicitly mark the trained columns and revise the conclusion that "RAW exceeds sRGB" to "RAW with one trained scalar parameter exceeds zero-shot sRGB.

- Linear-RAW failure mode. In Fig. 5 and elsewhere, low-bit linear-RAW often produces "I cannot see the object." This is a refusal, not a low-quality description, and aggregating it into mean scores conflates two regimes. Please report (1) the fraction of refusals per variant and (2) metric scores both with and without refusals included, so that the gap between linear and log+gamma can be attributed correctly.

- Fix the broken cross-reference. Sec. 5/Sec. F refers to "??" (e.g., "In ??, we summarize the statistics...")

- Resolve unit inconsistencies. Table 2 reports values like 0.188 (apparently fractions in [0,1]); Table 4 reports 30.3 (apparently percentages). Please use a single convention throughout the paper and indicate units in column headers.

- Define "AP95." Table 2 reports an "AP95" column that is not standard COCO terminology and not defined in the text (mAP is described as averaging IoU thresholds 0.5–0.95). If this is AP at IoU 0.95, please say so; if it is something else, please define.

- Fill in the Zurich rows of Table 1. Table 1 lists "-" for Zurich train/test images and number of classes. If Zurich is included in RAWDet-7, these counts are essential.

---

> ### Author Response · Authors · 2026-06-01
> **Response to Reviewer wFfa: Annotation Quality, Grounded-DINO Input Modality, and Confidence Threshold Sensitivity**
>
> We sincerely thank the Reviewer wFfa for their thorough and constructive feedback. The suggestions and comments have helped us identify important gaps and strengthen both the experimental evaluation and the clarity of the paper.
> We have now revised the current draft of the paper accordingly, with the revisions in text color blue.
> Below we have carefully addressed each concern and suggestion.
>
> *Provide absolute annotation quality. The current user study only establishes that the new annotations are preferred over the originals. Please include an absolute-quality assessment, e.g., expert manual correction of a random subset of (say) 200 images per source dataset, reporting precision/recall of the Grounded-DINO outputs against the corrected ground truth, broken down by class and by lighting condition. This is necessary because the entire benchmark rests on the assumption that 0.8-confidence Grounded-DINO outputs on RAW (or sRGB-of-RAW?) are reliable.*
>
> To assess absolute annotation quality, we manually inspected 80 images per source dataset, stratified across lighting conditions, yielding 480 images total. A single expert annotator recorded TP, FP, and FN for each class using IoU ≥ 0.5 as the match criterion, assessed by visual inspection. In the vast majority of TP cases, predicted bounding boxes exhibited clear, correct overlap with the target object. Per-class precision and recall, broken down by class and lighting condition, are reported in Fig. 9 in Appendix Section C. Across the 480 manually inspected images, the Grounded-DINO detector at a confidence threshold of 0.8 demonstrated consistently high annotation quality for the predominant object categories. Cars and persons, the two most frequent classes, achieved precision and recall above 0.95 in nearly all dataset--lighting combinations, confirming robust detection under both day and night conditions. Bicycles and motorcycles showed slightly more variability, with precision occasionally dropping to 0.75--0.88, primarily due to a small number of false positives rather than systematic misdetection. For Truck and Tram, we observe genuine scarcity in the data rather than detector failure. Excluding these, the macro-averaged precision and recall across all classes and conditions consistently exceeded 0.90, supporting the use of these annotations as a reliable ground truth for benchmarking purposes. These results confirm the reliability of 0.8-confidence Grounded-DINO outputs as annotation sources for the benchmark. We have added these details in Appendix C - Annotation Quality Validation.
>
> *Clarify whether Grounded-DINO was run on RAW or sRGB. Sec. 3.1 says "we re-annotated all datasets using Grounded-DINO 1.5," and elsewhere ("followed by human validation on the corresponding sRGB images") suggests the model was run on sRGB. Please state explicitly which input modality was given to the foundation model and how the resulting boxes were transferred back to the RAW-domain coordinates. If sRGB was used, please state which ISP / which sRGB version was used for each source dataset.*
>
> Grounded-DINO 1.5 was run exclusively on sRGB images. For all datasets except RAOD, the sRGB images used as input to the model were the images provided by the original dataset authors, processed through their respective ISP pipelines. For RAOD, no sRGB images were publicly available; we therefore processed the RAW images into sRGB using our own ISP pipeline, described in detail in Appendix Section G. Since sRGB and RAW images do not always share the same resolution, the resulting bounding boxes were transferred to RAW-domain coordinates by rescaling them according to the ratio between the RAW and sRGB image resolutions. We have added an explicit statement to Section 3.1 clarifying the input modality used for annotation and the coordinate transfer procedure.
>
> *Confidence threshold sensitivity and lighting bias. Foundation models trained on natural sRGB images will systematically have lower confidence on dark/night images (RAOD-Night, NOD), so a fixed 0.8 threshold may under-annotate night scenes. Please report (1) the distribution of confidence scores by source dataset and lighting condition, and (2) recall of new annotations versus the originals broken down by day/night. If there is a systematic skew toward day, this should be discussed as a limitation.*
>
> We have analysed the distribution of Grounded-DINO confidence scores broken down by lighting condition across RAOD. The mean confidence score was 0.899 for daytime images and 0.878 for night-time images, a difference of less than 0.02. The distributions overlap substantially, indicating that the fixed 0.8 threshold does not introduce a systematic day/night annotation bias in our data. The confidence score plots and precision--recall curves for day and night images are provided in Appendix Section D Figure 10, and Section H, Figures 14 and 15.

---

> ### Author Response · Authors · 2026-06-01
> **Response to Reviewer wFfa: Linear-RAW Failure Mode, Zurich Dataset Clarification, Minor Fixes**
>
> *Linear-RAW failure mode. In Fig. 5 and elsewhere, low-bit linear-RAW often produces "I cannot see the object." This is a refusal, not a low-quality description, and aggregating it into mean scores conflates two regimes. Please report (1) the fraction of refusals per variant and (2) metric scores both with and without refusals included, so that the gap between linear and log+gamma can be attributed correctly.*
>
> We have re-analysed our results separating refusals from scored responses. The refusal rates per variant are as follows: linear-RAW exhibits dramatically higher refusal rates across all bit depths, 10.58% at 4-bit, 7.72% at 6-bit, and 5.56% at 8-bit, whereas all non-linear variants (log, gamma, log+gamma, RGB-downsampled, sRGB) consistently fall below 1.6%, closely matching the sRGB baseline of approximately 1.8%, as shown in Table 5 in Appendix Section J.4. Figure 4 (main paper) and Figure 18 (appendix) show metric scores with and without refusals included, respectively. The scores for linear-RAW improve modestly after removing refusals, for example, Detail Level rises from 3.30 to 3.69 at 4-bit, and Detail Match from 2.87 to 3.02, but remain substantially below the corresponding log+gamma variants (4.32 and 5.03 at 4-bit). This confirms that the performance gap between linear-RAW and log+gamma has two compounding sources: a higher rate of outright refusals, and genuinely lower description quality in the non-refusal responses. The log+gamma transformation addresses both failure modes simultaneously. We have revised the paper to report refusal rates per variant explicitly (Table 5 in Appendix Section J.4) and discussed the analysis in Appendix Section J.
>
> *Fix the broken cross-reference. Sec. 5/Sec. F refers to "??" (e.g., "In ??, we summarize the statistics...")*
>
> The broken cross-reference has been corrected to Table 1 in the revised submission.
>
> *Resolve unit inconsistencies. Table 2 reports values like 0.188 (apparently fractions in [0,1]); Table 4 reports 30.3 (apparently percentages). Please use a single convention throughout the paper and indicate units in column headers.*
>
> All metric values are now expressed as decimals in the range [0, 1] consistently across all tables.
>
> *Define "AP95." Table 2 reports an "AP95" column that is not standard COCO terminology and not defined in the text (mAP is described as averaging IoU thresholds 0.5–0.95). If this is AP at IoU 0.95, please say so; if it is something else, please define.*
>
> AP95 refers to AP at a single IoU threshold of 0.95. We have added a definition in the text to clarify this.
>
> *Fill in the Zurich rows of Table 1. Table 1 lists "-" for Zurich train/test images and number of classes. If Zurich is included in RAWDet-7, these counts are essential.*
>
> Section 3.1 has been updated to clarify that while the Zurich dataset provides official train/test splits, RAWDet-7 retains only the full-resolution images for which both RAW and sRGB pairs were available, as the majority of the dataset consists of 488×488 patches randomly cropped from images (not provided otherwise as full scene images in the dataset). Additionally, since the Zurich dataset is used for RAW-to-sRGB conversion rather than object detection, no class annotations are available. Table 1 reflects the splits as provided in the original datasets, while the train and test splits used within RAWDet-7 are provided in Figure 16 of the appendix.
>
> Thank you for your valuable input.
> Please let us know if we have answered your questions.
> We are open for further discussions and look forward to your response.

---

> ### Author Response · Authors · 2026-06-01
> **Response to Reviewer wFfa: Circularity in the Description Protocol and Trained-vs-Zero-Shot Comparison**
>
> *Address the circularity in the protocol. Please rewrite the framing in Sec. 3.2, 5.3, and the Conclusion to make explicit that the metric measures agreement with sRGB-Gemini outputs, not absolute information preservation. The current language ("preserve substantially richer object-level information than naïve quantization") overclaims. A way to partially break the circularity would be to also include a setting where the ground truth is human-written descriptions on a smaller subset, and report whether the relative ordering of pipelines is the same.*
>
> Thank you for pointing this out. We agree that the previous wording could be read as claiming absolute information preservation. We have therefore revised Sec. 3.2, Sec. 5.3, and the Conclusion to state explicitly that the object-description metrics measure agreement with high-resolution sRGB-Gemini reference descriptions, not absolute semantic information preservation. More precisely, the protocol measures how much object-level information accessible to an RGB-pretrained VLM from high-resolution sRGB remains accessible after different RAW preprocessing and quantization pipelines.
>
> We also clarify that the protocol is not unvalidated. The high-resolution sRGB-Gemini descriptions define a fixed reference point for comparing different representations of the same scene under the same captioning model and set-of-marks protocol. We validate this reference and the judge protocol with human studies: in App. I.2, humans rate the sRGB reference descriptions at 4.07 ± 0.48 out of 5 over 1260 ratings; in App. I.5, humans score similarity, detail level, and detail match over 391 caption pairs, yielding positive correlations with Gemini-2.5-Flash-Lite judgments. Following the reviewer's suggestion to partially break the circularity, App. I.6 additionally reports an independent Claude Sonnet 4.6 judge evaluation, which preserves the same relative trends and correlates strongly with Gemini-2.5-Flash-Lite.
>
> Accordingly, we softened the framing from "information preservation" to "agreement with high-resolution sRGB reference descriptions" and "VLM-accessible object-level information." Under this explicitly defined and human-validated protocol, the empirical conclusion remains supported: linearly quantized RAW has substantially lower similarity, detail level, and detail match, while log and log+$\gamma$ processed RAW remain much closer to the high-resolution sRGB reference descriptions.
>
> *Clarify the trained-vs-zero-shot comparison in Table 2. The MM-Grounding-DINO results are presented as if they are like-for-like, but γ-scaling and log+γ involve training one scalar parameter while linear/log/sRGB are evaluated zero-shot. Please either (1) add a baseline where a single scalar gain is trained on top of sRGB inputs as well (so the comparison is fair), or (2) explicitly mark the trained columns and revise the conclusion that "RAW exceeds sRGB" to "RAW with one trained scalar parameter exceeds zero-shot sRGB."*
>
> Thank you for this suggestion. We have now rephrased the wording to remove the possible overclaim and discuss them as "on par results". We are currently running the experiment to address the asymmetry by scaling the sRGB inputs with a learnable gamma parameter, identical to the setup of training RAW inputs with gamma scaling. We will report the results as soon as we get them, and rephrase the observations accordingly.
>
> Thank you for your valuable input.
> Please let us know if we have answered your questions.
> We are open to further discussions and look forward to your response.

---

> ### Author Response · Authors · 2026-06-12
> **New MM-Grounding-DINO Results with sRGB + $\gamma$ Calibration**
>
> _Clarify the trained-vs-zero-shot comparison in Table 2. The MM-Grounding-DINO results are presented as if they are like-for-like, but γ-scaling and log+γ involve training one scalar parameter while linear/log/sRGB are evaluated zero-shot. Please either (1) add a baseline where a single scalar gain is trained on top of sRGB inputs as well (so the comparison is fair), or (2) explicitly mark the trained columns and revise the conclusion that "RAW exceeds sRGB" to "RAW with one trained scalar parameter exceeds zero-shot sRGB._
>
> We have now completed the requested experiment by training a single scalar $\gamma$ parameter on top of sRGB inputs, using the same frozen-MM-Grounding-DINO setup as for the RAW $\gamma$ and RAW Log + $\gamma$ variants. We have updated Table 2 accordingly by adding both sRGB + $\gamma$ and sRGB + Log + $\gamma$.
>
> The updated results show that 8-bit sRGB + $\gamma$ achieves the strongest overall performance. This is expected, since MM-Grounding-DINO is a foundation model pretrained on sRGB-like inputs, and optimizing even a single scalar parameter on sRGB further aligns the input statistics with the model's native modality. At the same time, the matched comparison gives a more nuanced and important result: 8-bit RAW + Log + $\gamma$ performs essentially on par with 8-bit sRGB + Log + $\gamma$ (0.202 vs. 0.204 mAP). We consider this the fairer comparison, since both inputs have the same bit depth and use the same lightweight one-parameter calibration. This supports our revised claim that, given suitable preprocessing, low-bit RAW remains fully competitive with sRGB even for a frozen detector strongly biased toward sRGB inputs.
>
> We have revised the wording in Section 5.2 and the Table 2 caption accordingly. We no longer frame the result as an unconditional RAW advantage over sRGB; instead, we emphasize that the requested symmetric comparison confirms the competitiveness of suitably scaled RAW under matched bit depth and matched lightweight adaptation.

---

### Review · Reviewer_tjgb · 2026-05-18

**Summary Of Contributions:**

This paper presents a dataset, RAWDet-7, that consists of the combination of four existing datasets of RAW images by consolidating the tasks and improving the annotations. The majority of instances consist of an object detection problem across seven object classes. The annotations were generated by a vision foundation model using standard RGB images. The paper then proceeds with experiments benchmarking the proposed dataset and evaluating the impact of different quantization schemes on the performance.

**Audience:**

No

**Audience Explanation:**

Without a proper quality assessment, I do not think this dataset is usable.

Since I am not familiar with the literature in CV for RAW images, I focus on the issues I see as a data paper rather than on RAW-related specificities, which I cannot properly judge.

**Claims And Evidence:**

No

**Claims Explanation:**

The paper describes a new dataset where the RAW images stem from existing datasets and the annotations are automatically generated using a standard RGB foundation model. Although a small-scale user study (100 images) that focuses on user "preference" is provided, I believe this does not suffice as an assessment of the quality of the new annotations.

The new annotations in this dataset are generated by a foundation model using standard RGB images. This in itself raises very serious concerns about both data quality and value added:
- A small-scale user study is described in the paper, in which users were asked about their “preference” between the old and the new annotations on 100 images. Although users reported preferring the new annotations ¾ of times, this does not qualify as a quality assessment, since it is not clear what “preference” means, and the sample is small and, potentially, not representative of corner cases.
- In addition, what is the value of using annotations generated via RGB images? These annotations come from standard RGB datasets and, thus, do not allow measuring any performance improvement with respect to using merely RGB.
- When comparing to one of the original datasets (RAOD), Fig 6 suggests a collapse in the precision of one of the classes (tram) for all levels of recall when using the new annotations. The authors suggest this due to the higher complexity of the new annotations, but this certainly calls for a deeper investigation, since it may be an indication of low-quality labels as well.

**Requested Changes:**

In order for this contribution to be potentially useful, a lot more effort is required for an in-depth quality assessment. This would require, at least:
- Creating a validation dataset that represents well all possible situations (sensors, conditions, scenes, object combinations, etc), which would probably correspond to several hundreds of a few thousand images.
- Producing high-quality human-annotated labels for this validation set that can be used to assess the quality of the generated annotations.
- Providing a detailed analysis of the quality of the provided annotations, with a focus on failure cases.

---

> ### Author Response · Authors · 2026-06-02
> **Response to Reviewer tjbg**
>
> We thank Reviewer tjbg for their detailed feedback on data quality. We have revised the paper substantially to address
> each point below.
>
> > _This paper presents a dataset, RAWDet-7, that consists of the combination of four existing
> > datasets of RAW images by consolidating the tasks and improving the annotations. The majority
> > of instances consist of an object detection problem across seven object classes. The annotations
> > were generated by a vision foundation model using standard RGB images. The paper then proceeds
> > with experiments benchmarking the proposed dataset and evaluating the impact of different
> > quantization schemes on the performance._
>
> We respectfully disagree with this summary. RAWDet-7 consolidates six RAW sources into a
> standardized benchmark with 24,864 training and 7,688 test images, seven unified object
> categories, corrected and denser annotations, coverage across sensors, bit depths, day/night/HDR
> conditions, controlled low-bit RAW quantization experiments, benchmarking across several
> detectors and MM-Grounding-DINO, and an object-description track. The contribution is broader
> than automatic relabeling: RAWDet-7 also introduces an object-description track to evaluate how
> RAW preprocessing and quantization pipelines affect agreement with high-resolution sRGB
> reference descriptions for object-level semantic, spatial, and contextual information.
>
> > _The paper describes a new dataset where the RAW images stem from existing datasets and the
> > annotations are automatically generated using a standard RGB foundation model. Although a
> > small-scale user study (100 images) that focuses on user "preference" is provided, I believe
> > this does not suffice as an assessment of the quality of the new annotations._
>
> The user study is not only "100 images" in the sense of 100 judgments. It used 100 randomly
> selected image pairs evaluated by 53 participants over 1325 pairwise comparisons. RAWDet-7
> annotations were preferred in 1002 cases, i.e., 75.62% of all comparisons. "Preference" means
> participants inspected two competing bounding-box annotations for the same image and selected
> the one they judged better. This study was designed as a relative comparison to the original
> annotations, which contain missing boxes, hallucinations, and incorrect labels.
>
> > _The new annotations in this dataset are generated by a foundation model using standard RGB
> > images. This in itself raises very serious concerns about both data quality and value added:_
>
> We respectfully disagree that model-assisted annotation is itself evidence against data quality
> or value added. Prior work has shown that large-scale vision datasets often contain missing,
> incomplete, or ambiguous human labels, which can make correct model predictions appear as false
> positives and underestimate model performance [R1, R2, R3]. Model-assisted relabeling can
> improve annotation coverage and evaluation fidelity, for example by converting ImageNet from
> single-label to localized multi-label supervision [R4] or by correcting false negatives in
> MS-COCO [R5]. The relevant question is therefore not whether annotations are human or automated,
> but whether the pipeline is accurate, exhaustive, consistent, and empirically validated.
>
> We validate these aspects at multiple levels. Appendix B provides a relative human validation
> with 53 participants and 1325 pairwise comparisons, where RAWDet-7 is preferred in 75.62% of
> cases. Appendix C complements this with an absolute expert audit over 480 images, recording TP,
> FP, and FN with class-wise precision and recall across dataset-lighting combinations in Fig. 9.
> Appendix G documents the ISP pipeline for RAOD, while Sec. 3.1 clarifies RAW-coordinate
> transfer. For the object-description track, Appendix J.2 validates reference descriptions with
> 63 participants and 1260 ratings, Appendix J.5 validates the Gemini-2.5-Flash-Lite judge against
> human judgments over 391 caption pairs, and Appendix J.6 confirms the same trends with an
> independent Claude Sonnet 4.6 judge.

---

> ### Author Response · Authors · 2026-06-02
> **Response to Reviewer tjbg**
>
> _A small-scale user study is described in the paper, in which users were asked about their “preference” between the old and the new annotations on 100 images. Although users reported preferring the new annotations ¾ of times, this does not qualify as a quality assessment, since it is not clear what “preference” means, and the sample is small and, potentially, not representative of corner cases._
>
> We agree that a preference study alone is not a complete quality assessment. We have therefore clarified the role of the user study and added an absolute annotation-quality audit in Appendix C. First, "preference" was not meant as an undefined subjective criterion: in Appendix B, participants were shown paired annotations for the same image, one from the original dataset and one from RAWDet-7, with randomized A/B ordering, and were asked to select the annotation they judged to be better after inspecting the bounding boxes. The setup and examples are shown in Appendix Fig. 8, with additional visual comparisons in Appendix Fig. 11. Moreover, the study should not be read as only 100 judgments: it used 100 randomly selected image pairs, but these were evaluated by 53 participants over 1325 pairwise comparisons, with RAWDet-7 preferred in 1002 cases, i.e., 75.62% of all comparisons.
>
> We nevertheless agree that this relative study alone does not fully address representativeness or corner cases. Therefore, we added an absolute annotation-quality audit in Appendix C, "Annotation Quality Validation". A single expert annotator manually inspected 80 images per source dataset, yielding 480 images total, stratified across lighting conditions. For each class, the annotator recorded TP, FP, and FN using IoU >= 0.5 as the matching criterion. This directly measures precision and recall rather than only user preference. The results are reported in Appendix Fig. 9. Cars and persons, the two dominant categories, achieve precision and recall above 0.95 in nearly all dataset-lighting combinations. Excluding tram, and bus where absent, macro-averaged precision and recall across classes and conditions consistently exceed 0.90. Thus, the revised paper now contains both a large relative human study and an absolute expert audit with class-wise precision/recall, stratified analysis, and failure-case discussion.
>
> _In addition, what is the value of using annotations generated via RGB images? These annotations come from standard RGB datasets and, thus, do not allow measuring any performance improvement with respect to using merely RGB._
>
> These are not standard RGB datasets. The source datasets are RAW datasets with corresponding sRGB views in most cases. Grounded-DINO 1.5 was run on sRGB images because it is an RGB foundation model; applying it directly to RAW sensor measurements would be an inappropriate use of the model and would artificially degrade annotation quality. For all datasets except RAOD, we used the sRGB images provided by the original dataset authors through their respective ISP pipelines. For RAOD, where no sRGB images were publicly available, we processed the RAW images into sRGB using our ISP pipeline, described in Appendix Section G. The resulting boxes were then transferred to RAW-domain coordinates by rescaling according to the RAW/sRGB resolution ratio. The annotations define scene-level object locations, and the benchmark evaluates detectors on RAW, low-bit quantized RAW, and sRGB representations using the same object-level ground truth. This is precisely what enables a controlled comparison of whether RAW and quantized RAW pipelines can match or exceed sRGB-based pipelines.

---

> ### Author Response · Authors · 2026-06-02
> **Response to Reviewer tjbg**
>
> _When comparing to one of the original datasets (RAOD), Fig 6 suggests a collapse in the precision of one of the classes (tram) for all levels of recall when using the new annotations. The authors suggest this due to the higher complexity of the new annotations, but this certainly calls for a deeper investigation, since it may be an indication of low-quality labels as well._
>
> We agree that the tram behavior in Sec. 5.4, Fig. 6 deserved a deeper analysis, and we have added this analysis in the revised manuscript. In Sec. 5.4, we now clarify that Fig. 6 compares RAOD under the "Old-Old", "Old-New", "New-New", and "New-Old" settings, and that the lower precision under the new annotations should not be interpreted directly as evidence of low-quality labels. We also provide separate RAOD-Day and RAOD-Night precision-recall curves in Appendix Fig. 14 and Appendix Fig. 15 to separate the effect by lighting condition.
>
> Most importantly, the added absolute annotation-quality audit in Appendix C, "Annotation Quality Validation", directly investigates this issue. In Appendix C, a single expert annotator manually inspected 80 images per source dataset, yielding 480 images total, stratified across lighting conditions, and recorded TP, FP, and FN for each class using IoU >= 0.5 as the match criterion. The resulting per-class precision and recall are reported in Appendix Fig. 9. This analysis shows that tram is a rare category in RAWDet-7, with zero true positives in four of the six dataset-lighting combinations. Thus, the apparent tram behavior is mainly explained by genuine data scarcity and class imbalance, rather than by a systematic annotation failure. By contrast, the frequent categories, especially car and person, achieve precision and recall above 0.95 in nearly all dataset-lighting combinations. Appendix H, Fig. 16 further shows the class imbalance and annotation differences between the original and RAWDet-7 annotations, including the fact that original RAOD contains hallucinated annotations for classes such as tram, truck, and car. We therefore revised the discussion to avoid over-interpreting a rare-class PR curve and to ground the analysis in the new class-wise and lighting-specific validation.
>
> _Would at least some individuals in TMLR's audience be interested in knowing the findings of this paper?: No_
>
> We respectfully disagree with this assessment. The conclusion appears to follow from an incomplete appreciation of the RAW-domain and low-bit sensing context, which the reviewer also acknowledges by stating that they are not familiar with the CV literature on RAW images. RAW-domain perception, quantization-aware sensing, learned ISP alternatives, sensor-to-task co-design, and efficient low-bit vision are active and growing topics within the broader machine learning and computer vision community. RAWDet-7 directly addresses a central bottleneck in this area: the lack of a large, standardized, multi-scenario benchmark for evaluating object detection and object-level description on RAW and quantized RAW inputs.
>
> This view is also supported by the other reviews. Reviewer thEa states that "The paper is relevant to RAW-domain perception, low-bit sensing, sensor-to-task co-design, and VLM evaluation, and the dataset itself fills a real gap, and large standardized RAW detection benchmarks remain scarce." Reviewer wFfa likewise states that "There is genuine community interest in (1) RAW-domain detection benchmarks, (2) sensor-to-task co-design and quantization-aware perception, and (3) how vision-language models behave on non-standard input regimes." These assessments align with our motivation: the benchmark is relevant to researchers working on robust perception, efficient vision, computational imaging, learned image processing, foundation models for detection, low-bit quantization, and dataset/benchmark construction.
>
> The benchmark consolidates six RAW sources across sensors, bit depths, and lighting conditions; provides seven standardized object categories; evaluates multiple detectors and MM-Grounding-DINO; studies controlled 4-bit, 6-bit, and 8-bit RAW quantization; and introduces an object-description track for measuring agreement with high-resolution sRGB reference descriptions. These findings are not limited to a narrow dataset-release contribution, but speak to the broader question of whether machine perception pipelines can benefit from RAW or low-bit RAW inputs instead of relying only on conventional sRGB ISP outputs.

---

> ### Author Response · Authors · 2026-06-02
> **Response to Reviewer tjbg**
>
> _Without a proper quality assessment, I do not think this dataset is usable._
>
> The revised version now includes both relative and absolute quality assessment. The relative study covers 1325 human pairwise comparisons by 53 participants, while the new absolute audit in Appendix C covers 480 manually inspected images across all source datasets and lighting conditions, with expert TP/FP/FN labels and class-wise precision/recall reported in Appendix Fig. 9. These additions directly support the usability of RAWDet-7 as a benchmark.
>
> _Since I am not familiar with the literature in CV for RAW images, I focus on the issues I see as a data paper rather than on RAW-related specificities, which I cannot properly judge._
>
> We appreciate the reviewer's focus on dataset quality and openness on the lack of domain knowledge. However, several concerns raised in the review are tied to RAW-specific benchmarking practice, especially the use of sRGB views of the same RAW scenes for annotation and the transfer of boxes back to RAW coordinates. We have therefore strengthened the dataset validation independently of RAW-specific assumptions: the revised paper now provides a clearer annotation protocol in Sec. 3.1, an explicit sRGB-to-RAW coordinate transfer description in Sec. 3.1, a human preference study in Appendix B, an expert precision/recall audit in Appendix C, and a failure-case analysis in Appendix C and Sec. 5.4. These address the core data-quality concerns directly.
>
> _In order for this contribution to be potentially useful, a lot more effort is required for an in-depth quality assessment. This would require, at least:_
>
> We agree that annotation quality is critical, and the revised version adds the requested in-depth quality assessment. Appendix C, "Annotation Quality Validation", goes beyond the original preference study and provides direct TP/FP/FN accounting, class-wise precision and recall in Appendix Fig. 9, and analysis across source datasets and lighting conditions.
>
> _Creating a validation dataset that represents well all possible situations (sensors, conditions, scenes, object combinations, etc), which would probably correspond to several hundreds of a few thousand images._
>
> We have added a validation study in Appendix C with 480 manually inspected images, covering all source datasets and stratified across lighting conditions. This provides several hundred images across sensors, datasets, and conditions, while remaining feasible given the cost of dense object-level inspection.
>
> _Producing high-quality human-annotated labels for this validation set that can be used to assess the quality of the generated annotations._
>
> The added validation study in Appendix C provides expert manual labels for the inspected validation subset. Specifically, a single expert annotator inspected 80 images per source dataset, yielding 480 images total, stratified across lighting conditions, and recorded TP, FP, and FN for each class using IoU >= 0.5 as the matching criterion. These expert annotations are then used in Appendix Fig. 9 to compute class-wise precision and recall for each dataset-lighting combination. This directly assesses both false positives and false negatives in the generated annotations, rather than relying only on relative user preference. The resulting analysis shows high precision and recall for the dominant classes, especially car and person, and identifies rare-class limitations such as tram separately.

---

> ### Author Response · Authors · 2026-06-02
> **Response to Reviewer tjbg**
>
> _Providing a detailed analysis of the quality of the provided annotations, with a focus on failure cases._
>
> We now provide a detailed class- and lighting-specific analysis in Appendix C, including rare-class failure cases. The main observed failure mode is not systematic annotation failure, but low instance counts for rare classes such as tram, which strongly affects class-wise precision-recall interpretation. This is also discussed in Sec. 5.4, and the RAOD-specific day/night precision-recall curves are provided in Appendix Fig. 14 and Appendix Fig. 15.
>
> Overall, the revised manuscript directly addresses the reviewer's data-quality concerns. RAWDet-7 is supported by a large relative human study, a new absolute expert audit over 480 images, explicit precision/recall analysis, clarified sRGB-to-RAW annotation transfer, and a failure-case analysis. The evidence, therefore, supports the use of RAWDet-7 as a reliable benchmark for RAW-domain object detection and low-bit RAW quantization.
>
> We hope that these clarifications and the additional experiments help the reviewer reassess the initial evaluation of the work. We appreciate the opportunity to clarify these points and would be happy to answer any further questions. We also hope the reviewer will consider that the other reviews independently recognize the relevance and usefulness of the benchmark: Reviewer thEa writes that "Pairing denser detection labels with an object-description track is a genuinely new angle for RAW-domain evaluation, and the user studies do real work in supporting the claims", while Reviewer wFfa writes that "a consolidated benchmark with corrected labels and a standardized protocol for low-bit evaluation is a useful resource." We believe the revised manuscript now directly addresses the concerns raised here while preserving the central contribution of RAWDet-7 as a timely and useful benchmark for RAW-domain perception.
>
> Please let us know if we have answered your questions.
> We are open to further discussions and look forward to your response.
>
> References
>
> [R1] Tsipras, D., Santurkar, S., Engstrom, L., Ilyas, A. and Madry, A., 2020. From ImageNet to image classification: Contextualizing progress on benchmarks. In International Conference on Machine Learning, pp. 9625-9635. PMLR.
>
> [R2] Xu, M., Bai, Y. and Ghanem, B., 2019. Missing labels in object detection. In Proceedings of the IEEE/CVF Conference on Computer Vision and Pattern Recognition Workshops, pp. 1-10.
>
> [R3] Suri, S., Rambhatla, S., Chellappa, R. and Shrivastava, A., 2023. SparseDet: Improving sparsely annotated object detection with pseudo-positive mining. In Proceedings of the IEEE/CVF International Conference on Computer Vision, pp. 6770-6781.
>
> [R4] Yun, S., Oh, S.J., Heo, B., Han, D., Choe, J. and Chun, S., 2021. Re-labeling ImageNet: From single to multi-labels, from global to localized labels. In Proceedings of the IEEE/CVF Conference on Computer Vision and Pattern Recognition, pp. 2340-2350.
>
> [R5] Chun, S., Kim, W., Park, S., Chang, M. and Oh, S.J., 2022. ECCV caption: Correcting false negatives by collecting machine-and-human-verified image-caption associations for MS-COCO. In European Conference on Computer Vision, pp. 1-19. Springer.

---

### Review · Reviewer_thEa · 2026-05-18

**Summary Of Contributions:**

The paper introduces RAWDet-7, a benchmark for object detection and object description on quantized RAW images. It consolidates four RAW datasets (PASCAL RAW, Zurich, NOD-Nikon/Sony, RAOD) into ~25k train and 7.6k test images, re-annotates them with Grounded-DINO 1.5 plus human validation, and unifies labels into seven MS-COCO-style categories. Beyond detection, the authors add an object-description track using set-of-marks prompting with Gemini-2.5-Pro on high-resolution sRGB references. Faster R-CNN, PAA, RetinaNet, and MM-Grounding-DINO are benchmarked at 4/6/8-bit with linear, log, learnable gamma, and log+gamma scaling.

The paper has cleaner and denser annotations (preferred in 75.62% of 1325 pairwise comparisons), a new description track validated by humans (4.07/5), and a consistent finding that log/gamma scaling recovers most of the gap to sRGB at low bit depths. There are some main weakness: the "RAW can surpass sRGB" claim rests on comparisons where RAW pipelines have trained parameters and sRGB does not, and Gemini is used both to generate and to judge descriptions.

**Additional Comments:**

This is a solid benchmark contribution and I lean toward acceptance after revision. Pairing denser detection labels with an object-description track is a genuinely new angle for RAW-domain evaluation, and the user studies do real work in supporting the claims. My main asks are to soften the sRGB-vs-RAW framing until the comparison is symmetric, and to be transparent about the Gemini-judges-Gemini setup. With clearer annotation provenance, more visible per-class breakdowns, and slightly more cautious wording on the VLM results, the paper would be a useful addition to TMLR.

**Audience:**

Yes

**Audience Explanation:**

Yes. The paper is relevant to RAW-domain perception, low-bit sensing, sensor-to-task co-design, and VLM evaluation, and the dataset itself fills a real gap, and large standardized RAW detection benchmarks remain scarce.

**Broader Impact Concerns:**

No major concerns.

**Claims And Evidence:**

Yes

**Claims Explanation:**

The core claims are reasonably backed. The construction pipeline is clearly described, prior RAW datasets are compared in Tab. 1, and the user studies (1325 detection pairs, 1260 caption ratings, 391 caption-pair ratings for LLM-as-judge correlation) give the dataset claims real grounding. The quantization trend is also consistent across architectures in Fig. 2 and at the VLM level in Tab. 2 and Fig. 4.

**Requested Changes:**

Make the RAW-vs-sRGB comparison symmetric in Tab. 2, or flag the asymmetry clearly. Ideally add an sRGB row with equally lightweight calibration (a per-channel scale or one learned gamma on sRGB) so the "RAW beats sRGB at 6-bit" claim is not driven by adaptation alone. For Fig. 2, confirm in text that resolution, augmentation, and training budget are identical between RAW and sRGB.

Tighten the annotation description. Sec. 3.1 and App. A.2 say predictions above 0.8 were kept after manual inspection of "a large subset" plus human validation. Please clarify whether every retained box was checked or only a sample, whether humans ever added missed boxes, and how false negatives from the 0.8 threshold were handled.

Address the Gemini-judges-Gemini issue in Sec. 5.3. A small additional run with a different judge (e.g. GPT-4o or Claude) on a subset, or a direct human comparison between linear and log+gamma at one bit depth, would significantly strengthen the central claim of the description track.

Add per-dataset and per-class breakdowns to the main paper. The dataset spans 10/12/14/24-bit sensors and is highly imbalanced across classes (Tab. 3, Fig. 13). Fig. 7 moves in this direction but only for Faster R-CNN; more complete tables in the appendix would clarify where the benchmark is reliable.

Clarify how "quantized RAW" here relates to real low-bit sensor capture — post-hoc quantization of higher-bit RAW does not reproduce real ADC behaviour or photon-limited noise. One or two sentences in Sec. 4 would suffice.

Minor: "RegeX"/"BLeU" in Sec. 3.2 should be Regex Match / BLEU. "Her,e" in the Fig. 3 caption. A broken cross-reference "In ??," appears in App. F. Tab. 1 lists Zurich with no train/test split or class count even though it contributes to the totals.

---

> ### Author Response · Authors · 2026-06-01
> **# Response to Reviewer thEa: Annotation Transparency, Independent Judge Validation, and Per-Dataset Breakdown Clarifications**
>
> We sincerely thank the Reviewer thEa for their thorough and constructive feedback. Their comments have helped us identify important gaps and strengthen both the experimental evaluation and the clarity of the paper.
> We have now revised the current draft of the paper accordingly, with the revisions in text color blue.
> Below we have carefully addressed each concern and suggestion.
>
> *Tighten the annotation description. Sec. 3.1 and App. A.2 say predictions above 0.8 were kept after manual inspection of "a large subset" plus human validation. Please clarify whether every retained box was checked or only a sample, whether humans ever added missed boxes, and how false negatives from the 0.8 threshold were handled.*
>
> The 0.8 confidence threshold was selected after manual inspection of annotation quality over a large subset of RAWDet-7, and further validated through a user study with 53 participants and 1325 pairwise comparisons, in which RAWDet-7 annotations were preferred over the original dataset annotations in 75.62% of cases (as mentioned in sec 3.1 of the main paper); Given there are a total 24,864 images with total 251,168 annotations, individual box-level verification of the full dataset was not performed. In Appendix B, we show the study in which human validators confirmed annotation quality but did not add missed boxes. To directly address the concern about false negatives introduced by the 0.8 threshold, we have added a new annotation quality study in Appendix C - Annotation Quality Validation, in which a single expert annotator recorded TP, FP, and FN for 80 images per source dataset (480 images total), stratified across lighting conditions. For the predominant categories (cars and persons), precision and recall both exceed 0.95 in nearly all datasets--lighting combinations, confirming that the threshold does not introduce a systematic false-negative problem; lower recall for rare classes such as tram reflects genuine data scarcity rather than threshold-induced suppression.
>
> *Address the Gemini-judges-Gemini issue in Sec. 5.3. A small additional run with a different judge (e.g. GPT-4o or Claude) on a subset, or a direct human comparison between linear and log+gamma at one bit depth, would significantly strengthen the central claim of the description track.*
>
> To address the concern that the object-description track uses Gemini-generated reference descriptions together with a Gemini-family judge, we additionally evaluate the same caption outputs with Claude Sonnet 4.6. We keep the evaluated captions, image variants, scoring prompts, and aggregation protocol fixed, and change only the judge model. As shown in Fig. 20 in the appendix, the scores from Claude Sonnet 4.6 are strongly aligned with those from Gemini-2.5-Flash-Lite, with correlations of $\rho=0.90$ for Similarity, $\rho=0.70$ for Detail Level, and $\rho=0.91$ for Detail Match. Importantly, the relative trends across image representations are preserved: descriptions from log+$\gamma$ processed RAW images remain consistently closer to the high-resolution sRGB reference descriptions than descriptions from linearly quantized low-bit RAW images. This independent-judge check does not treat the sRGB reference descriptions as absolute semantic ground truth, but it shows that the reported ordering of RAW processing pipelines is not specific to using a Gemini-family model as judge. Therefore, we use Gemini-2.5-Flash-Lite as the primary judge for the full evaluation due to its lower cost at scale, while the Claude Sonnet 4.6 results provide model-family-independent support for the conclusions of the object-description track. We have updated Section J.6 in the appendix to include these details.
>
> *Add per-dataset and per-class breakdowns to the main paper. The dataset spans 10/12/14/24-bit sensors and is highly imbalanced across classes (Tab. 3, Fig. 13). Fig. 7 moves in this direction but only for Faster R-CNN; more complete tables in the appendix would clarify where the benchmark is reliable.*
>
>
> Per-dataset and per-class breakdowns are already provided in the appendix: Figures 12 and 13 report per-dataset detection performance for Faster R-CNN and PAA respectively, Figure 14 and 15 shows precision-recall curves for RAOD broken down by class comparing old and new annotations for day and night images respectively, and Figure 16 presents class distribution statistics across all sub-datasets. In the revised verison, we have added explicit pointers to these figures from the main paper to ensure readers are directed to these details. Given the page limit, we are unable to find the space to fit these in the main paper, however, if the reviewer suggests to move certain sections of the current version main paper to the appendix to make space, especially for Figure 16, then we are happy to do so.

---

> ### Author Response · Authors · 2026-06-01
> **Response to Reviewer thEa: Quantization Limitations, Minor Corrections, and RAW-vs-sRGB Symmetry Evaluation**
>
> *Clarify how "quantized RAW" here relates to real low-bit sensor capture — post-hoc quantization of higher-bit RAW does not reproduce real ADC behaviour or photon-limited noise. One or two sentences in Sec. 4 would suffice.*
>
> We acknowledge that post-hoc quantization of higher-bit RAW data does not fully replicate the behaviour of a real low-bit sensor, as genuine low-bit capture introduces ADC-specific noise characteristics and photon-limited shot noise at a level not present in a digitally quantized high-bit signal. Our quantization procedure is therefore best understood as a controlled simulation of reduced bit-depth representation rather than a faithful emulation of real low-bit sensor hardware. We have added an explanation to Section 4 making this distinction explicit and noting it as a limitation of the current benchmark.
>
> *Minor: "RegeX"/"BLeU" in Sec. 3.2 should be Regex Match / BLEU. "Her,e" in the Fig. 3 caption. A broken cross-reference "In ??," appears in App. F. Tab. 1 lists Zurich with no train/test split or class count even though it contributes to the totals.*
>
> All four issues have been corrected in the revised submission. "RegeX" and "BLeU" have been standardised to "Regex Match" and "BLEU" respectively. The typo "Her,e" in the Figure 3 caption and the broken cross-reference in Appendix F have both been fixed. Section 3.1 has been updated to clarify that while the Zurich dataset provides official train/test splits,p. Additionally, since the Zurich dataset is used for RAW-to-sRGB conversion rather than object detection, no class annotations are available. Table 1 reflects the splits as provided in the original datasets, while the train and test splits used within RAWDet-7 are provided in Figure 16 of the appendix.
>
> *Make the RAW-vs-sRGB comparison symmetric in Tab. 2, or flag the asymmetry clearly. Ideally add an sRGB row with equally lightweight calibration (a per-channel scale or one learned gamma on sRGB) so the "RAW beats sRGB at 6-bit" claim is not driven by adaptation alone. For Fig. 2, confirm in text that resolution, augmentation, and training budget are identical between RAW and sRGB*
>
> Thank you for this suggestion. We have now rephrased the wordings to remove the possible overclaim. We are currently running the experiment to address the asymmetry by scaling the sRGB inputs with a learnable gamma parameter, identical to the setup of training RAW-inputs with gamma scaling. We will report the results as soon as we get them, and rephrase the observations accordingly.
>
> All training conditions are identical across RAW and sRGB inputs, including augmentation and training budget. The resolution of RAW and sRGB images is the same for RAOD, Nikon, and Sony, but differs slightly for Pascal RAW and Zurich, identical to the original image resolutions from the respective datasets. For all RAW experiments, the R, G, G, B Bayer channels are extracted and the two green channels are averaged to produce a three-channel representation, which results in half the effective spatial resolution of the original RAW input and sRGB. For Pascal RAW and Zurich, where RAW and sRGB resolutions already differ, the resulting resolution varies accordingly. We have added these details in Sec. 5 of the main paper.
>
> Thank you for your valuable input.
> Please let us know if we have answered your questions.
> We are open to further discussions and look forward to your response.

---

> ### Author Response · Authors · 2026-06-12
> **New MM-Grounding-DINO Results with sRGB + $\gamma$ Calibration**
>
> _Make the RAW-vs-sRGB comparison symmetric in Tab. 2, or flag the asymmetry clearly. Ideally add an sRGB row with equally lightweight calibration (a per-channel scale or one learned gamma on sRGB) so the "RAW beats sRGB at 6-bit" claim is not driven by adaptation alone. For Fig. 2, confirm in text that resolution, augmentation, and training budget are identical between RAW and sRGB_
>
> We have now completed the requested experiment by training a single scalar $\gamma$ parameter on top of sRGB inputs, using the same frozen-MM-Grounding-DINO setup as for the RAW $\gamma$ and RAW Log + $\gamma$ variants. We have updated Table 2 accordingly by adding both sRGB + $\gamma$ and sRGB + Log + $\gamma$.
>
> The updated results show that 8-bit sRGB + $\gamma$ achieves the strongest overall performance. This is expected, since MM-Grounding-DINO is a foundation model pretrained on sRGB-like inputs, and optimizing even a single scalar parameter on sRGB further aligns the input statistics with the model's native modality. At the same time, the matched comparison gives a more nuanced and important result: 8-bit RAW + Log + $\gamma$ performs essentially on par with 8-bit sRGB + Log + $\gamma$ (0.202 vs. 0.204 mAP). We consider this the fairer comparison, since both inputs have the same bit depth and use the same lightweight one-parameter calibration. This supports our revised claim that, given suitable preprocessing, low-bit RAW remains fully competitive with sRGB even for a frozen detector strongly biased toward sRGB inputs.
>
> We have revised the wording in Section 5.2 and the Table 2 caption accordingly. We no longer frame the result as an unconditional RAW advantage over sRGB; instead, we emphasize that the requested symmetric comparison confirms the competitiveness of suitably scaled RAW under matched bit depth and matched lightweight adaptation.

---

### Decision · Action_Editor_vBjM · 2026-07-07

**Recommendation:** Accept with minor revision

**Additional Comments:**

The final decision on this manuscript was made after carefully reviewing the comments of the three reviewers, the authors' response and the revised text. The reviewers' opinions are divided: two recommend acceptance, praising the practical utility and the extensive revisions made by the authors, while one maintains rejection, raising concerns about the scientific rigour of the data construction.

The AE recognises that the paper makes a meaningful contribution to the community by standardising and integrating multiple raw datasets, and by introducing practical benchmarks such as low-bit quantisation and object description tracks. The authors' constructive efforts during the rebuttal, such as adding the expert audit and cross-validation, are highly commendable.

However, the reviewer who recommended rejection raised a valid point: generating ground truth (GT) annotations from a single sRGB view discards the rich dynamic range inherent in RAW data. This introduces a contradiction when evaluating RAW-based prediction models (for example, RAW models might detect valid objects hidden in sRGB shadows but be penalised as false positives). The concern regarding confirmation bias during the post hoc expert audit of model-assisted labels is also justified.

While these limitations do not completely diminish the practical value of the benchmark, they cannot be left ambiguous. Therefore, this paper is accepted, under the strict condition that the authors implement the following requirements in their camera-ready version:

- Explicit discussion of RAW limitations: In a dedicated 'Limitations' section, the authors must explicitly and transparently discuss how, because the GT annotations were derived from standard sRGB views via foundation models, they may fail to capture objects that are only visible under RAW's high dynamic range (HDR) conditions.

- Acknowledgement of annotation bias: the authors must add an objective discussion acknowledging the potential 'confirmation bias' inherent in post hoc human audits of model-assisted annotations (i.e. the risk of humans overlooking the same objects missed by the model).

- Incorporate related work: As suggested by the reviewers, the authors must properly integrate and discuss literature closely related to RAW object detection and image restoration.

**Audience:**

Yes

**Audience Explanation:**

Reviewers wFfa and thEa affirm that, given the lack of large-scale, standardized RAW detection benchmarks, this study serves as a useful resource for bridging gaps in RAW perception and VLM evaluation, and that it has garnered significant interest within the community. On the other hand, reviewer tjgb points out that, while avoiding RAW-specific debates, the dataset will not be adopted by readers unless appropriate quality assessments are conducted. Overall, provided that quality is assured, this is an area of interest to many readers.

**Claims And Evidence:**

Yes

**Claims Explanation:**

Two reviewers assessed that the claims were sufficiently supported based on a clear construction pipeline, multifaceted annotation validation using user surveys and LLMs, and extensive experimental results. On the other hand, one reviewer sharply pointed out concerns regarding data quality, such as the fact that automatic annotation using an sRGB base model omits the wide dynamic range information specific to RAW images, as well as insufficient validation and reduced accuracy in certain classes. In their rebuttal, the authors added 480 expert audits and cross-model validations, and by qualifying their claims, they obtained consensus from wFfa and thEa that “there is sufficient evidence to support the main claims.” Technical limitations, such as the “inconsistency of evaluating RAW predictions using sRGB-based ground truth” pointed out by tjgb, still remain. Therefore, the acceptance is conditional on these limitations being clearly stated in the “Limitations” section of the camera-ready manuscript.

---

> ### Author Response · Authors · 2026-07-22
> **Summary of Revisions**
>
> We thank the Action Editor and reviewers for the constructive feedback and the conditional acceptance. All additions and changes in the current manuscript are highlighted in olive green for the convenience of the Action Editor and reviewers. We have addressed all three required revisions as follows:
> 1. Explicit discussion of RAW limitations. We have extended the Limitations section (end of the paper) by transparently stating that annotations derived from standard sRGB views may fail to capture objects that are only visible under the high dynamic range of RAW data (e.g., objects hidden in sRGB shadows or highlights), and that RAW-based models may consequently be penalized as false positives for detecting valid objects absent from the sRGB-derived ground truth.
> 2. Acknowledgement of annotation bias. At the end of Section 3.1 and in Sec. C of the Appendix, we have added an objective discussion acknowledging the potential confirmation bias inherent in post-hoc human audits of model-assisted annotations, i.e., the risk that human auditors overlook the same objects missed by the annotation model.
> 3. Incorporation of related work. We have revised the Related Work section to properly integrate and discuss literature closely related to RAW object detection and image restoration, including RAW-domain detection methods and a dedicated discussion of RAW image restoration, and we clarify how our benchmark is orthogonal to and complements these lines of work.
>
> We sincerely thank the Action Editor and the reviewers once again for their time, thoughtful feedback, and guidance, which have considerably improved the quality and clarity of the paper. We kindly invite the Action Editor to review the current draft and would be grateful for any further suggestions, so that we can promptly upload the final camera-ready version.